# The evolution of division of labour in structured and unstructured groups

Guy Alexander Cooper[1,2]*, Hadleigh Frost[3], Ming Liu[2], Stuart Andrew West[2]

[1]St John's College, Oxford, United Kingdom; [2]Department of Zoology, University of Oxford, Oxford, United Kingdom; [3]Mathematical Institute, University of Oxford, Oxford, United Kingdom

**Abstract** Recent theory has overturned the assumption that accelerating returns from individual specialisation are required to favour the evolution of division of labour. Yanni et al., 2020, showed that topologically constrained groups, where cells cooperate with only direct neighbours such as for filaments or branching growths, can evolve a reproductive division of labour even with diminishing returns from individual specialisation. We develop a conceptual framework and specific models to investigate the factors that can favour the initial evolution of reproductive division of labour. We find that selection for division of labour in topologically constrained groups: (1) is not a single mechanism to favour division of labour—depending upon details of the group structure, division of labour can be favoured for different reasons; (2) always involves an efficiency benefit at the level of group fitness; and (3) requires a mechanism of coordination to determine which individuals perform which tasks. Given that such coordination must evolve prior to or concurrently with division of labour, this could limit the extent to which topological constraints favoured the initial evolution of division of labour. We conclude by suggesting experimental designs that could determine why division of labour is favoured in the natural world.

## Editor's evaluation

This manuscript presents a theoretical study of the evolution of division of labor, exploring the impact of topology, the convexity and concavity of fitness returns on investment, and different biological modes through which division of labor may arise. This is a difficult topic to study as division of labor evolved long ago, and many theoretical predictions have proven difficult to directly test. The results presented here may provide the next step necessary to produce truly testable hypotheses on how division of labor evolves, and will be of interest to evolutionary biologists, mathematical biologists, and biophysicists.

*For correspondence:
guy.cooper@zoo.ox.ac.uk

## Introduction

Division of labour, where cooperating individuals specialise to carry out distinct tasks, plays a key role at all levels of biology (*Bourke, 2011*; *Queller, 1997*; *Maynard Smith and Szathmáry, 1995*; *West et al., 2015*). Cells are built by genes carrying out different functions (*Bourke, 2011*; *Levin and West, 2017*). In clonal groups of bacteria, cells specialise to produce and secrete different factors that facilitate growth (*Dragoš et al., 2018a*; *Veening et al., 2008*; *West and Cooper, 2016*). Pathogens rely on division of labour for protection from the host immune response and competitors (*Ackermann et al., 2008*; *Diard et al., 2013*). Multicellular organisms are composed of reproductive germ cells and sterile somatic cells that are not passed to the next generation (*Bourke, 2011*; *Maynard Smith and Szathmáry, 1995*). The ecological dominance of the social insects arises from division of

**Figure 1.** Division of labour is favoured by accelerating returns from individual specialisation. (**A**) Theory has shown that either a linear or diminishing return from more cooperation (or reproduction) favours uniform cooperation, with all individuals investing the same amount of effort into cooperation and reproduction (i.e. no division of labour) (*Cooper and West, 2018*; *Michod, 2006*; *Schiessl et al., 2019*). (**B**) In contrast, an accelerating return from more cooperation (or reproduction) favours reproductive division of labour, with some individuals specialising in high levels of cooperation (helpers) and others in low levels of cooperation (reproductives) (*Cooper and West, 2018*; *Michod, 2006*; *Schiessl et al., 2019*).

labour between queens and the different types of workers (castes) (*Hölldobler and Wilson, 1990*; *Oster and Wilson, 1978*).

It has long been established that the evolution of division of labour requires an efficiency benefit from individual specialisation (*Figure 1A and B* for reproductive division of labour) (*Bourke, 2011*; *Cooper and West, 2018*; *Ispolatov et al., 2012*; *Michod, 2006*; *Oster and Wilson, 1978*; *Schiessl et al., 2019*; *Biggart, 1776*; *Maynard Smith and Szathmáry, 1995*; *Solari et al., 2013*). In particular, that there is an accelerating (convex) return when individuals commit more effort to a particular task, such that twice the investment more than doubles the return (*Bourke, 2011*; *Cooper and West, 2018*; *Ispolatov et al., 2012*; *Michod, 2006*; *Solari et al., 2013*). An accelerating return from individual investment can exist for several reasons. A task could become more effective as more effort is put into it, or it could be carried out with diminishing costs. This could occur if there are large upfront costs from performing a task. For instance, any reproduction by a cell in Volvocine groups first requires individual growth to the size of a daughter colony (*Michod, 2006*). Alternatively, there could be a disruptive cost to carrying out multiple tasks at the same time if the tasks do not mix well. For instance, in cyanobacteria the enzymes that fix environmental nitrogen are degraded by oxygen, a bi-product of photosynthesis (*Flores and Herrero, 2010*).

In contrast, Yanni et al. found that division of labour between helpers and reproductives can sometimes be favoured even when there are diminishing (concave) returns from individual specialisation (*Yanni et al., 2020*). Specifically, reproductive division of labour could arise in topologically constrained groups—where each cell in a spatially structured group shares cooperative benefits with only their direct neighbours (*Staps and Tarnita, 2020*; *Yanni et al., 2020*). Their analyses suggested that this is particularly likely to occur in sparsely structured groups, where cells have a small number of neighbours (*Yanni et al., 2020*). This is a novel result. Diminishing returns means that specialised individuals are inefficient, and earlier work suggested that division of labour could not be favoured in this situation (*Figure 1A*, *Cooper and West, 2018*; *Michod, 2006*; *Schiessl et al., 2019*). Consequently, this result has the potential to overturn our understanding of the factors that favour the evolution of division of labour.

However, there are several issues that still need to be resolved with how topological constraints can favour division of labour. Why exactly do the predictions of this new theory differ from previous theory? Are special group structures the only way to alter the predictions of the previous theory, or is this an example of a more general phenomenon (*Rueffler et al., 2012*)? Do these findings rely upon implicit assumptions, which may not be reasonable during the initial evolution of division of labour? Answering these questions is not only of theoretical importance: it is also key for planning future empirical studies. Quantifying the shape of the returns from individual specialisation has been assumed to be a fundamental step in determining why division of labour was favoured in some species, but not others (*Diard et al., 2013*; *Dragoš et al., 2018a*; *Flores and Herrero, 2010*; *Koufopanou, 1994*; *Mridha and Kummerli, 2021*; *Strassmann et al., 2000*; *Veening et al., 2008*).

We first use the methodology developed by *Rueffler et al., 2012*, to derive the general conditions that favour the initial evolution of reproductive division of labour between helpers and reproductives.

We then use this framework to examine when and why topological constraints can favour division of labour. More specifically, we determine the ultimate cause of division of labour in specific topologically constrained groups, such as filaments and branching growths, as well as in a general analysis of arbitrary group structures. We then ask whether division of labour without an accelerating return from individual specialisation could arise in groups without topological constraints. To test our hypothesis that between-individual coordination is required for division of labour in these cases, we re-examine our models while assuming that cells adopt helper and reproductive roles randomly (no coordination). We finish by suggesting experimental designs for determining why division of labour has evolved in specific species.

## Results and discussion

### General invasion analysis

We follow previous studies by assuming that individual fitness is the product of individual viability, which is the chance of surviving to maturity, and individual fecundity, which is proportional to the number of offspring if the individual reaches maturity (*Cooper and West, 2018*; *Michod, 2006*; *Yanni et al., 2020*). We examine the specific case of reproductive division of labour between helpers and reproductives, where helpers are more cooperative, contributing to a higher viability for group members, and reproductives are less cooperative, contributing to higher individual fecundity.

We consider an initial population of clonal groups each containing $n$ individuals, in which all individuals cooperate at the evolutionarily stable (ES) level ($z^*$), which is the level that cannot be outcompeted by a mutant strain that uses a different level of uniform cooperation across the group (*Maynard Smith, 1982*). We then ask when this population of uniform cooperators can be invaded by a mutant strain that employs a reproductive division of labour.

Without loss of generality, we assume that the mutant strain is composed of $n_h$ helpers that invest $z_h \geq 0$ into cooperation and $n_r$ reproductives that invest $z_r \geq 0$ into cooperation (where $n_h + n_r = n > 2$ and $z_h > z_r$). We set individual fitness as the product of individual fecundity, $F > 0$, and individual viability, where helpers and reproductives may in principle have different viability functions, $V_h > 0$ and $V_r > 0$ (but see Appendix C.2) (*Michod, 2006*; *Yanni et al., 2020*). The fitness of the clonal group is given by the sum of individual fitness:

$$W(z_h, z_r) = n_h F(z_h) V_h(z_h, z_r) + n_r F(z_r) V_r(z_h, z_r), \tag{1}$$

where the first term on the right-hand side is the total fitness of the prospective helpers and the second term on the right-hand side is the total fitness of the prospective reproductives.

Fecundity is determined by an individual's investment in cooperation ($F = F(z)$, where $z$ is the focal individual's level of cooperation) and viability is determined by the level of cooperation at the level of the group ($V_h = V_h(z_h, z_r)$ and $V_r = V_r(z_h, z_r)$). We assume that there is a tradeoff between fecundity and viability such that higher individual cooperation leads to lower individual fecundity ($F'(z) < 0$), but that more cooperation leads to a higher viability for all individuals (i.e. $V_h^{z_h}, V_h^{z_r}, V_r^{z_h}, V_r^{z_r} > 0$, where superscripts denote partial derivatives). We assume that viability selection occurs just prior to reproduction. This is consistent with previous models and ensures that there there is no feedback between a cell's viability and its ability to produce cooperative benefits for the group (*Cooper and West, 2018*; *Michod, 2006*; *Yanni et al., 2020*).

We determine the invadability conditions that favour reproductive division of labour by applying the general approach of *Rueffler et al., 2012*. The key step is to approximate the relative fitness of a reproductive division of labour mutant by taking a second-order Taylor expansion of fitness, centred on the resident strategy of uniform cooperation, $z^*$:

$$W\left(z^* + \Delta z_h, \; z^* + \Delta z_r\right) - W\left(z^*, \; z^*\right) \approx$$

$$\underbrace{W^{z_h} \Delta z_h}_{2(a)} + \underbrace{W^{z_r} \Delta z_r}_{2(b)} + \underbrace{\tfrac{1}{2} W^{z_h z_h} \Delta z_h^2}_{2(c)} + \underbrace{\tfrac{1}{2} W^{z_r z_r} \Delta z_r^2}_{2(d)} + \underbrace{W^{z_h z_r} \Delta z_h \Delta z_r}_{2(e)}, \tag{2}$$

where $\Delta z_h > \Delta z_r$ captures the change in the level of cooperation for mutant helpers and reproductives, respectively, which we assume are small in magnitude. The superscripts represent first- and second-order partial derivatives, where all partial derivatives are evaluated at the resident strategy

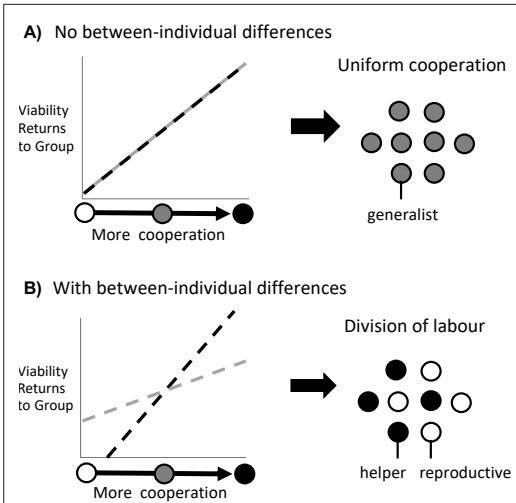

**Figure 2.** Division of labour is favoured by between-individual differences. Division of labour is favoured if some individuals are predisposed to being reproductives or helpers. (**A**) In the absence of another mechanism, if there are no differences between individuals (black and grey lines), then division of labour is not favoured. (**B**) If some individuals can produce larger viability benefits than others (black line), or if some individuals can access greater fecundity benefits than others (grey line), then this predisposition favours division of labour.

of uniform cooperation ($z_h = z_r = z^*$). If a mutant strain exists such that **Equation 2** is positive ($W(z^* + \Delta z_h, z^* + \Delta z_r) > W(z^*, z^*)$), then division of labour between helpers and reproductives is favoured to evolve. Conversely, if for all possible mutant strains, **Equation 2** is negative ($W(z^*, z^*) > W(z^* + \Delta z_h, z^* + \Delta z_r)$), then uniform cooperation is evolutionarily stable.

## The three pathways to division of labour

We found that reproductive division of labour could be favoured for three distinct reasons, corresponding to different subsets of terms on the right-hand side of **Equation 2**. Our results for reproductive division of labour, where fitness is partitioned as the product of fecundity and viability, align with those found by **Rueffler et al., 2012**, for division of labour more generally. We now go through these three distinct scenarios.

### Scenario 1: Accelerating returns from individual specialisation

The first and most studied scenario that can favour division of labour is when there are accelerating returns from individual specialisation. This occurs if there is an accelerating fitness return from either helper specialisation in cooperation or reproductive specialisation in fecundity (**Figure 1**; **Cooper and West, 2018**; **Michod, 2006**; **Oster and Wilson, 1978**).

Mathematically, this scenario is a consequence of the third and fourth terms of the Taylor expansion (2$c$ and 2$d$), which capture the second-order fitness effect of a small, unilateral change in cooperation by either prospective helpers or reproductives, respectively. Division of labour is favoured to evolve whenever at least one of 2$c$ and 2$d$ is greater than zero ($W^{z_h z_h} > 0$ or $W^{z_r z_r} > 0$), and where we have assumed that the first two terms are both zero ($W^{z_h} = 0$ and $W^{z_r} = 0$; see between-individual differences below; **Figure 2A**). In either scenario, an efficiency benefit to group fitness arises from individual specialisation because the more effort that each individual puts into a task, the better they can perform that task. Rueffler et al. termed these kinds of scenario as 'accelerating performance functions' (**Rueffler et al., 2012**).

### Scenario 2: Between-individual differences

The second scenario that can favour reproductive division of labour is when there are pre-existing differences between individuals in the group, such that some individuals are predisposed to one task or the other. For example, if some individuals can secure larger viability benefits for the group at the same fecundity cost as others (**Figure 2B**).

This scenario is captured by the first two terms of the Taylor expansion (2$a$ and 2$b$), which are the first-order fitness effects from a small, unilateral change in the level of cooperation by prospective helpers or reproductives. If the direct fitness effects are non-zero (positive or negative) at the resident strategy of uniform cooperation ($W^{z_h} \neq 0$ or $W^{z_r} \neq 0$), then division of labour can invade independently of any higher-order effects (the remaining terms in **Equation 2**).

We term this scenario 'between-individual differences' because it requires that there is pre-existing phenotypic or environmental variation between individuals in the group. For the within-species case, ancestral groups are usually composed of clonal or highly related individuals, who will be phenotypically similar or identical. Consequently, this mechanism could be less important for the division of labour except when there are consistent differences in the microenvironment experienced by different

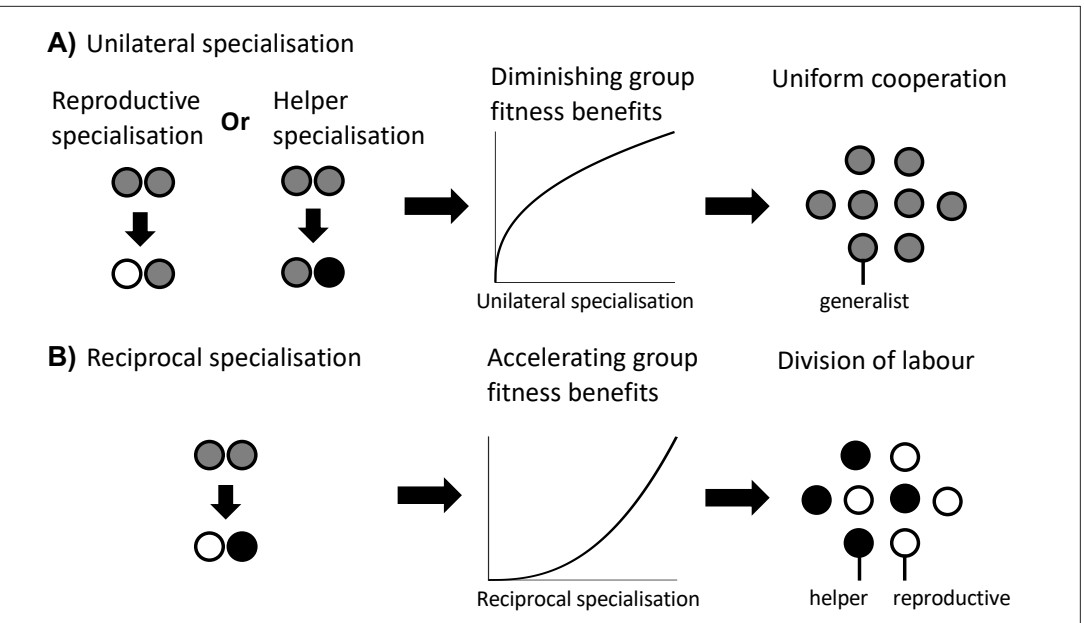

**Figure 3.** Division of labour is favoured by reciprocal specialisation. We assume that there are diminishing returns from specialisation in either viability or fecundity (*Figure 1A*). (**A**) In this case, a unilateral increase in cooperation by helpers or a unilateral decrease in cooperation by reproductives leads to a diminishing fitness benefits to the group, which favours uniform cooperation (no division of labour). (**B**) In contrast, a reciprocal increase in cooperation by helpers (more viability benefits provided by helpers) and a decrease in cooperation by reproductives (larger reproductive fecundity) can produce an accelerating return to the fitness of the group if the benefits of increased cooperation are preferentially directed to reproductives. Thus, reciprocal specialisation can still favour division of labour, even though the returns from individual specialisation are diminishing. In the middle plots of (**A**) and (**B**), only the shape of the benefits from increased specialisation is plotted.

individuals (*Tverskoi et al., 2018*; *Tverskoi and Gavrilets, 2021*). In contrast, this scenario is likely to be widespread in the evolution of non-reproductive division of labour between species, such as for mutualisms or symbioses (*Kiers et al., 2011*; *Rueffler et al., 2012*; *Wyatt et al., 2014*). Individuals of different species often differ in their abilities to perform certain tasks (*Kiers et al., 2011*). Rueffler et al. termed this scenario 'positional', but we avoid that term to prevent confusion with topological position (*Rueffler et al., 2012*).

Between-individual differences provide a first-order fitness benefit to dividing labour, and so it does not matter whether the subsequent benefits of increased cooperation or fecundity are accelerating or diminishing (*Figure 1*), so long as these benefits are different for different individuals (*Figure 2*). When some individuals are predisposed to being either helpers or reproductives, then individual specialisation provides an efficiency benefit to group fitness by capitalising on these inherent differences.

## Scenario 3: Reciprocal specialisation

The final scenario that can favour division of labour is when reciprocal specialisation by both helpers and reproductives provides a fitness benefit to the group (*Figure 3*). This scenario requires two key conditions. First, simultaneous specialisation, where some individuals invest more in cooperation (more viability benefits for the group), and others invest less in cooperation (greater individual fecundity; but see below). Second, this reciprocal specialisation must provide a group-level fitness benefit, because the increased benefits of cooperation are preferentially directed towards reproductives.

Mathematically, this scenario involves the last term of the Taylor expansion ($2e$; $W^{z_h z_r} \Delta z_h \Delta z_r$). This term is generated by a between-individual, second-order fitness effect, capturing how increased investment in viability by some individuals affects the returns from increased investment in fecundity by others, and vice versa. Rueffler et al. referred to this as a 'synergistic benefit' to division of labour (*Queller, 1985*; *Queller, 2011*; *Rueffler et al., 2012*).

Critically, this scenario still involves an efficiency benefit to specialisation, but at the level of group fitness rather than in each fitness component separately (Appendix C.1). By this we mean that there is an accelerating fitness benefit to the group when helpers and reproductives reciprocally specialise, leading to a higher group fitness than in groups with uniform cooperation (generalists). This occurs if the increased help given to reproductives is sufficiently amplified by the increased fecundity of reproductives (*Yanni et al., 2020*). This synergistic efficiency benefit can favour division of labour even if there are diminishing returns from individual specialisation.

Division of labour by reciprocal specialisation can also evolve without a joint mutation in the level of cooperation of both helpers and reproductives (no simultaneous specialisation). In this case, the chance invasion (to fixation) of a slightly deleterious mutant that specialises in only one phenotype can destabilise uniform cooperation, creating a selection pressure for the other phenotype to also specialise that is greater than the selection pressure to purge the initial mutant. In this two-step scenario, it is nevertheless the synergistic benefit from reciprocal specialisation that makes division of labour more efficient.

## Group structure in the general framework

Our above analysis has shown that reproductive division of labour can be favoured for three reasons: (1) accelerating returns make individual specialisation more efficient; (2) between-individual differences make individual specialisation more efficient; and (3) there is a synergistic efficiency benefit from reciprocal specialisation. These results agree with previous analyses by *Rueffler et al., 2012*.

We now use this framework to examine how and why topological constraints can favour division of labour in the absence of an accelerating return from individual specialisation (i.e. when scenario 1 does not hold). We ask three questions. First, can topological constraints favour division of labour by between-individual differences (scenario 2), and/or by reciprocal specialisation (scenario 3)? Second, are topological constraints the only way to evolve a division of labour without an accelerating return from individual specialisation? Third, does the evolution of division of labour by between-individual differences (scenario 2) and reciprocal specialisation (scenario 3) require coordination between individuals to determine which cells become helpers or reproductives?

### Question 1: How do topological constraints favour division of labour?

We consider two spatial models, based on the group structures proposed by Yanni et al., to examine whether topologically constrained groups favour division of labour by: (a) between-individual differences and/or (b) reciprocal specialisation (*Yanni et al., 2020*).

### Can topological constrains lead to division of labour by between-individual differences?

Consider a group in which cells alternately have either two or three neighbours, in a branching structure (*Figure 4*). Such a group structure might have occurred for some early forms of multicellular life (*Yanni et al., 2020*). We term cells with three neighbours 'node' cells and cells with two neighbours 'edge' cells. We assume that cells investing an amount $z \geq 0$ into cooperation produce an amount $H(z)$ of a public good. We assume non-accelerating returns from individual specialisation (i.e. $H''(z) \leq 0$ or $F''(z) \leq 0$). The cell keeps a fraction $1 - \lambda$ of the public good that it produces, and the remaining fraction $\lambda$ is shared equally between its direct neighbours (the 'shareability' of cooperation: $0 < \lambda \leq 1$). We assume that the viability of a cell is equal to the sum of the public good that it absorbs.

For this model, we find that for all social traits ($\lambda > 0$), reproductive division of labour by between-individual differences can evolve (*Figure 4A*; Appendix A.1). This occurs because different cells have different viability-fecundity tradeoffs depending on their position in the group. Edge cells receive relatively less public good from their (fewer) neighbours, and so pay a smaller opportunity cost from decreased fecundity (increased cooperation). In contrast, node cells receive relatively more public good from their (more numerous) neighbours, and so pay a larger opportunity cost from decreased fecundity (increased cooperation). Consequently, this between-cell difference favours node cells to specialise in fecundity (reproductives) and edge cells to specialise in increased cooperation (helpers). Importantly, because this pathway to division of labour is driven entirely by a first-order effect (2a and 2b), it does not require a second-order efficiency benefit from specialisation (2c, 2d, or 2e).

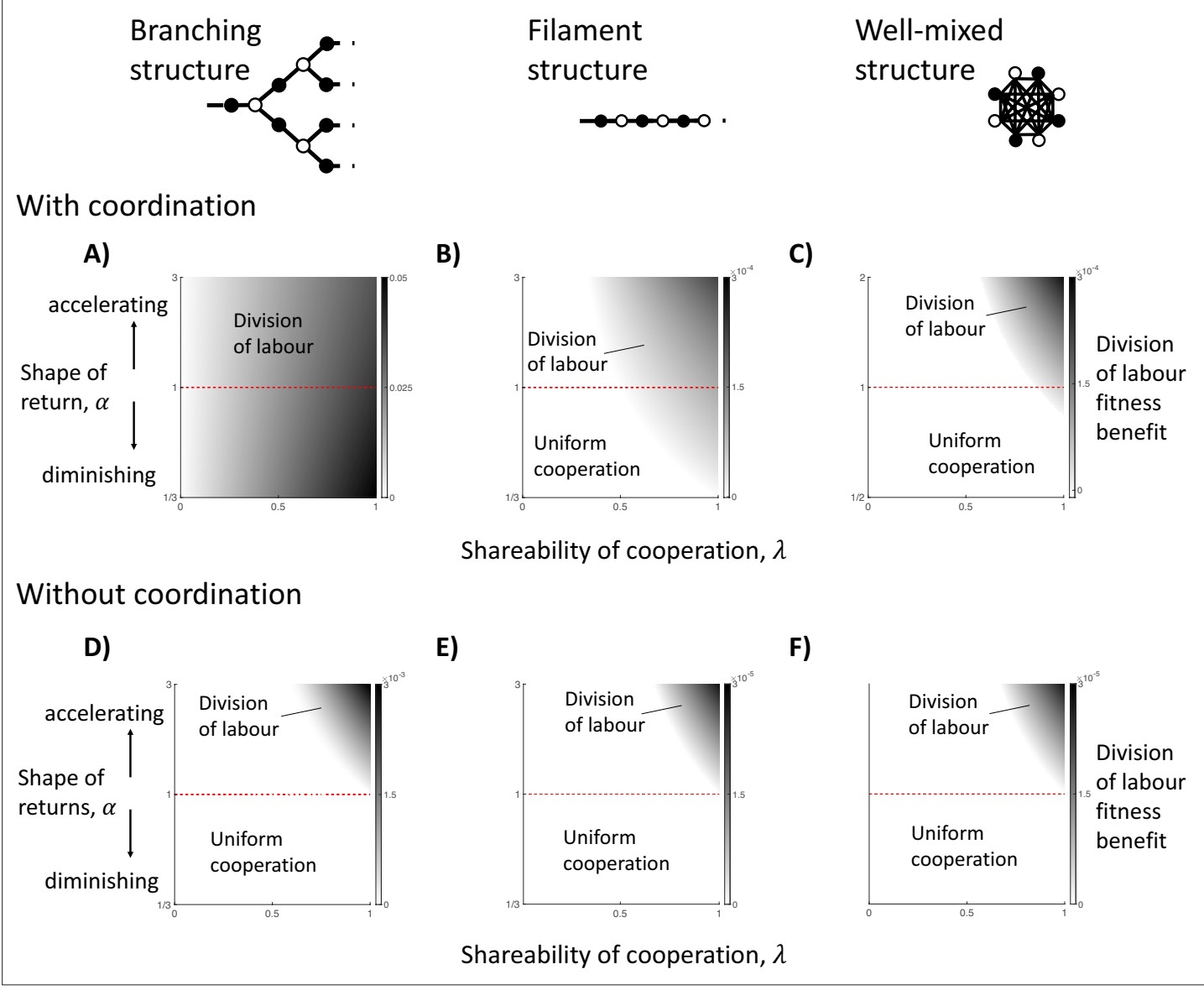

**Figure 4.** The impact of topological constraints on the division of labour. We show here different scenarios in which division of labour can evolve (non-white shades) and the size of its fitness benefit if so (darker shades). We consider three specific spatial models, including: a branching structure (A and D); a filament structure (B and E); and a well-mixed group (C and F). We consider when cells know their location in the group when specialising (with coordination; A–C) and when they do not (without coordination; D–F), in which case cells specialise randomly. (A) In a branching group structure with coordination, division of labour with diminishing returns from specialisation ($\alpha < 1$) can be favoured by between-individual differences whenever the benefits of cooperation are shared ($\lambda > 0$). (B) In a filament structure with coordination, division of labour with diminishing returns from specialisation ($\alpha < 1$) can be favoured by reciprocal specialisation when cells share a sufficient majority of the public good they produce with neighbours (e.g. when linear returns: $\lambda > \frac{1}{2}$). (C) In a well-mixed group with coordination, division of labour with diminishing returns from specialisation ($\alpha < 1$) can be favoured by reciprocal specialisation if cells share an even larger proportion of the public good they produce with neighbours (e.g. when linear returns: $\lambda > \frac{n-1}{n}$). (D–F) When cells specialise randomly (no coordination) across all three spatial models, then division of labour can only evolve if there is an accelerating return from specialisation ($\alpha > 1$). Throughout, we have assumed a linear return from fecundity specialisation, $F(x) = 1 - x$, and allow for a non-linear return from investment in cooperation, $H(x) = x^{\alpha}$, where $\alpha$ controls the shape of the return.

More generally, a formal analysis of arbitrary group structures reveals that division of labour by between-individual differences can always evolve whenever the number of neighbours varies for different cells in the group (Appendix A.2).

## Can topological constraints lead to division of labour by reciprocal specialisation?

Consider a one-dimensional chain of cells, as examined by Yanni et al. Chains are found in species like cyanobacteria that form filaments of cells, and such structures might have been important at the onset of the evolution of multicellularity (*Figure 4*, *Yanni et al., 2020*). We assume arbitrarily that 'odd' cells along the filament are putative helpers and 'even' cells are putative reproductives. We otherwise make the same assumptions as for the branching structure model: there is a non-accelerating return from individual specialisation (i.e. $H''(z) \leq 0$ or $F''(z) \leq 0$), and the cell keeps a fraction $1 - \lambda$ of the public good that it produces, with the remaining fraction $\lambda$ being shared equally by its direct neighbours.

If the amount of public good shared with neighbours is sufficiently large (high $\lambda$), then we find that division of labour via reciprocal specialisation can evolve (*Figure 4B*; Appendix A. 3). For instance, in the case of linear fecundity and public good returns ($H''(z) = F''(z) = 0$), division of labour by reciprocal specialisation can evolve if helpers share more of the public good that they produce with their neighbours than they keep for themselves ($\lambda > \frac{1}{2}$). If there are diminishing returns from specialisation ($H''(z) < 0$ or $F''(z) < 0$), then division of labour can still be favoured but then the amount of the public good preferentially shared with neighbours must be even greater still (higher $\lambda$; *Figure 4E*).

For an arbitrary group structure, our analysis in the previous section implies that division of labour can evolve by between-individual differences, unless every cell in the group has the same number of neighbours. Consider a group in which every cell has exactly d neighbours. In this case, we show (Appendix A.5) that division of labour can still be favoured due to reciprocal specialisation if:

$$\lambda\mu > d \tag{3}$$

$\lambda$ is the shareability of cooperation as defined previously and $\mu$ captures how easily the group can be 'bi-partitioned'. That is, $\mu$ is a measure of the extent that the group can be divided into two classes of cells such that cells are neighbours with many cells of the opposing class and few neighbours of their own class. Thus, reciprocal specialisation can favour division of labour if: (1) groups are more sparse (low $d$); (2) groups are structured such that helpers can be neighbours with reproductives more than with other helpers, and vice versa (high $\mu$); and/or (3) when the benefits of cooperation are preferentially shared with neighbours (high $\lambda$). In combination, these three factors amplify the synergistic benefits of reciprocal helper and reproductive specialisation, which can produce an accelerating fitness return for the group, even when there are non-accelerating returns from individual specialisation (Appendix C.1).

If one or two of these factors are particularly favourable for reciprocal specialisation, then the condition(s) on the remaining factor(s) can be relaxed. For instance, cells in a filament have only two neighbours ($d = 2$), and the potential alternation of helpers and reproductives in the filament means that helpers can share their cooperative public goods with reproductives exclusively (maximal $\mu$). Consequently, reciprocal specialisation is possible even when the shareability of cooperation is reasonably low (e.g. $\lambda > \frac{1}{2}$ for linear benefits).

To conclude, topological constraints are not a single explanation for division of labour, in that they can favour division of labour for two different biological reasons. Different group structures can lead to either between-individual differences favouring division of labour (scenario 2) or reciprocal specialisation favouring division of labour (scenario 3). In all cases, there is an efficiency benefit from specialisation at the level of group fitness even if the returns from individual specialisation are non-accelerating.

## Question 2: Are topological constraints required for division of labour without accelerating returns from individual specialisation?

We considered a well-mixed social group of $n$ cells, where all cells share the benefits of cooperation with one another, and so there are no topological constraints (*Figure 4*). We then examined whether division of labour could be favoured by: (a) between-individual differences; and/or (b) reciprocal specialisation. In both cases, we assume that when a cell invests $z$ into cooperation, it produces

an amount $H(z)$ of a public good. A cell keeps a fraction $1 - \lambda$ of the public good that it produces and the remaining fraction $\lambda$ is shared by the rest of the social group members equally. We again consider the case where there is a non-accelerating return from individual specialisation ($H(z)'' \leq 0$; $F''(z) \leq 0$).

## Can between-individual differences favour division of labour without a topological constraint?

In the well-mixed group of identical cells, we find that division of labour cannot arise by between-individual differences (Appendix A.4). This is because all cells have the same number of neighbours, which we have shown more generally can never produce between-individual differences. This prediction could be violated if one of our assumptions do not hold: for instance, if there are consistent differences in the microenvironment that predispose some cells to one task or the other (**Tverskoi et al., 2018**; **Tverskoi and Gavrilets, 2021**; **Yanni et al., 2020**).

## Can reciprocal specialisation favour division of labour without a topological constraint?

In the well-mixed group of identical cells, if the amount of public good shared with neighbours is sufficiently large (high $\lambda$), then we find that division of labour via reciprocal specialisation can evolve (**Figure 4F**; Appendix A.4). If there are linear returns from increased specialisation ($H''(z) = F''(z) = 0$), then division of labour can evolve when the public good produced by an individual benefits an average group member more than the producer ($\lambda > \frac{n-1}{n}$; **Figure 4C**). These results are like those found for a filament of cells (**Figure 4B**). In both cases, more generous sharing (higher $\lambda$) means that the synergistic benefits of reciprocal specialisation can be great enough to compensate for the non-accelerating returns from individual specialisation. In well-mixed groups, very generous sharing ($\lambda \approx 1$) also compensates for the fact that helpers are neighbours with all other helpers (no sparsity and minimally 'bi-partionable').

To conclude, the well-mixed model shows that a topological constraint is not required for the evolution of division of labour with non-accelerating returns from individual specialisation. This result is in direct contradiction to that of **Yanni et al., 2020**. This difference arises because helpers in their model always benefit at least as much as any of its neighbours from its own public good production ($\lambda \leq \frac{n-1}{n}$) (**Yanni et al., 2020**). Our model allows for biological scenarios where the public good benefits an average neighbour more than the producer ($\lambda > \frac{n-1}{n}$). For instance, reproductive cells in cyanobacteria may absorb more of the fixed nitrogen produced by helpers than helpers do to meet the large energetic requirements of cell duplication and division (**Flores and Herrero, 2010**; **Herrero et al., 2016**; **Meeks and Elhai, 2002**). At the extreme, the public good can be an 'others-only' trait that benefits neighbours but not the producer at all ($\lambda = 1$) (**Pepper, 2000**). An example of this are the dispersal benefits provided by stalk cells in *Dyctiostelium discodeum* fruiting bodies or the self-sacrificing behaviour of helper cells in *Salmonella enterica* infections (**Ackermann et al., 2008**; **Strmecki et al., 2005**). Consequently, our model allows for a wider spectrum of biologically realistic scenarios. Critically, division of labour can be favoured in a group of well-mixed cells because it provides an efficiency benefit at the group level, via reciprocal specialisation (scenario 3), in an analogous way to our model with a one-dimensional chain of cells (question 1b).

## Question 3: Is coordination required to favour division of labour without accelerating returns from individual specialisation?

We hypothesised that the benefits of between-individual differences (scenario 2) or reciprocal specialisation (scenario 3) rely on the implicit assumption that cells are coordinating which individuals specialise to become reproductive and helpers. This matters because mechanisms for coordinating division of labour, such as between cell signalling, might not be expected to exist before division of labour has evolved (**Cooper et al., 2022**; **Liu et al., 2021**). Consequently, if coordination was required, then this could limit the extent to which topological constrains favour the initial evolution of division of labour.

We investigated this hypothesis by repeating our above analyses, while assuming that cells do not have access to information that allows them to coordinate their phenotypes. Specifically, cells do not know if they are 'odd' or 'even', or if they are 'edge' or 'node'. We assumed instead that a reproductive

division of labour mutant induces each cell in the group to adopt the role of a helper or reproductive with a uniform probability (random specialisation). Random specialisation has been observed in a number of microbes (*Ackermann et al., 2008*; *Diard et al., 2013*; *Veening et al., 2008*). For filaments, branching group structures, and well-mixed groups, we found that division of labour can no longer evolve with non-accelerating returns from individual specialisation (*Figure 4D–E*; Appendices B.1 and B.2). In Appendix B.3, we have shown that this result holds for any group structure.

Consequently, for division of labour to evolve with non-accelerating returns from individual specialisation, there must exist some mechanism to coordinate which cells specialise to perform which tasks. It is possible that the mechanism need not produce a perfect allocation of labour across the group as analysed in our models (*Liu et al., 2021*). However, because division of labour cannot be favoured to evolve if role allocation is fully random, at least some degree of even imperfect between-cell coordination will be required.

A clear example of coordinated division of labour in topologically constrained groups is the use of between-cell signalling in some cyanobacteria filaments to determine which cells become sterile nitrogen fixing heterocysts and which cells become reproductive photosynthesisers (*Flores and Herrero, 2010*; *Meeks and Elhai, 2002*). However, a signal to coordinate distinct phenotypes must exist prior to or concurrently with the emergence of division of labour, and so a topological constraint is less likely to have favoured the initial evolution of division of labour in cyanobacteria. Alternatively, division of labour could have been favoured by an accelerating return from individual specialisation (scenario 1), with coordination only being favoured to evolve subsequently. Empirically, an accelerating return seems likely, as the key tasks performed by reproductives and helpers do not mix well (photosynthesis and nitrogen fixation) (*Flores and Herrero, 2010*; *Meeks and Elhai, 2002*).

These analyses do not suggest that topological constraints could never favour the initial evolution of division of labour. For instance, a pre-existing cue could allow division of labour to initially evolve with a metabolically cheaper form of coordination. More specifically, phenotype could be determined in response to the number of neighbours or the local concentration of some resource. Further, if there are pre-existing differences between individuals due to a pre-existing mechanism of coordination, then this mechanism can be co-opted to coordinate division of labour. However, the biological plausibility of any pre-existing mechanism would need to be explicitly justified and modelled on a case-by-case basis (*Duarte et al., 2011*). This would include modelling the metabolic cost, benefits and effectiveness of the mechanism (*Cooper et al., 2022*; *Duarte et al., 2012*; *Liu et al., 2021*). Empirically, while between-cell coordination has evolved in several labour-dividing microbial species, further studies—such as ancestral-state reconstructions—are needed to show whether coordination evolved prior to, concurrently with, or subsequent to division of labour in individual species.

In contrast, an accelerating return from individual specialisation depends on non-adaptive factors such as the physics, chemistry, or external constraints associated with the public good and its production. For instance, an accelerating return can arise if some intermediate products associated with cooperation and fecundity do not 'mix-well' on a chemical level. Consequently, no additional adaptive or pre-adaptation argument is needed to explain this pathway to division of labour.

## Distinguishing the ultimate causes of division of labour in the wild

How can we distinguish empirically which of the different scenarios favoured real-world examples of division of labour (*Figure 5*)? We suggest experimental designs for microbial systems, where mixtures of helper and reproductive cells are grown together, and which make use of methods to genetically manipulate and measure the relative levels of cooperation and reproduction of each phenotype (*Ackermann et al., 2008*; *Diard et al., 2013*; *Dragoš et al., 2018a*; *Dragoš et al., 2018b*; *Mavridou et al., 2018*; *Mridha and Kummerli, 2021*; *van Gestel et al., 2015*). These are rough suggestions for the kind of experiments required, as details and possibilities will vary system from system, depending upon factors such as the degree of specialisation, the mechanism by which labour is divided, and what manipulations are possible. In addition, these experiments would need to follow from key first steps, such as demonstrating division of labour and a tradeoff between reproduction and cooperation (*Diard et al., 2013*; *Dragoš et al., 2018a*; *Dragoš et al., 2018b*; *Veening et al., 2008*; *Zhang et al., 2020*; *Figure 5*).

Testing for accelerating returns from individual specialisation (scenario 1): In at least three treatments, vary the level of cooperation performed by the helpers (*Figure 5A* top-left), to test whether the

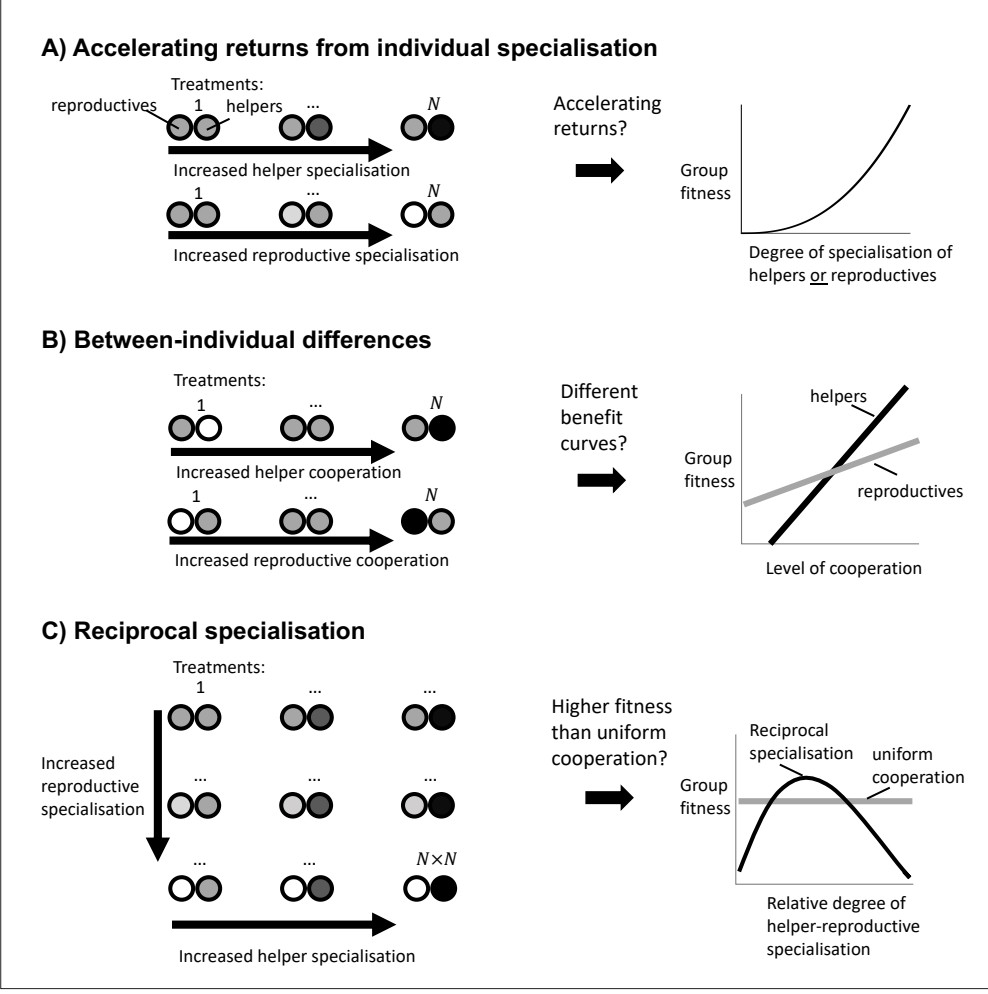

**Figure 5.** Experimental guidelines to distinguish the causes of division of labour. (**A**) To test whether division of labour is favoured by an accelerating return from individual specialisation, we must separately determine whether an increase in helper cooperation or a decrease in reproductive cooperation leads to an accelerating increase in group fitness. (**B**) To test whether division of labour is favoured by between-individual differences, we must determine whether an increase in cooperation by helpers produces a different group fitness benefit than an increase in cooperation by reproductives. (**C**) To test whether division of labour is favoured by reciprocal specialisation, we must determine whether there exists at least one relative degree of helper-to-reproductive specialisation for which group fitness is greater than the fitness of uniform cooperation.

benefits of increased cooperation are accelerating (**Figure 5A** right). Across at least three other treatments, vary the level of reproduction by the reproductives (**Figure 5A** bottom-left), to test whether the benefits of increased fecundity are accelerating (**Figure 5A** right). At least three treatments are required to be able to test for non-linear (accelerating) benefits.

Testing for between-individual differences (scenario 2): In some treatments, vary the level of cooperation of helpers (**Figure 5B** top-left). In other treatments vary the level of cooperation of reproductives (**Figure 5B** bottom-left). If division of labour evolves in this system by between-individual differences, then we should observe that group fitness varies differently depending on whether it is helpers or reproductives that are cooperating at a higher rate (**Figure 5B** right).

Testing for reciprocal specialisation (scenario 3): Use a classic $N \times N$ factorial experiment where the level of cooperation performed by helpers and the level of reproduction performed by reproductives are both varied (**Figure 5C** left). Division of labour is favoured by reciprocal specialisation if there is a significant interaction term between these two factors, with at least one treatment that produces a group fitness larger than that for uniformly cooperating cells (**Figure 5C** right).

## Conclusion

Division of labour can be favoured to evolve without accelerating returns from individual specialisation. Nevertheless, for this to occur requires: (a) between-individual differences in task-efficiency or synergistic benefits from reciprocal specialisation and (b) a mechanism to coordinate which individuals perform which tasks. In contrast, accelerating returns can favour division of labour without a mechanism to coordinate task allocation, possibly making it more likely to favour the initial evolution of division of labour. Ultimately, determining the relative importance of these different pathways to division of labour is an empirical question, requiring experimental studies of the type we have outlined above.

## Materials and methods

### Resident strategy of uniform cooperation

We start by solving for the ESS strategy where both types of individuals invest the same amount in cooperation ($z_h = z_r = z$; uniform cooperation). This is the level of uniform investment in cooperation, $z^*$, for which there is no selection for a uniform change in the amount of cooperation by all individuals in the group:

$$\frac{\partial W(z,z)}{\partial z}|_{z=z^*} = 0.$$

More explicitly, we can write this as:

$$n_h F V_h \left( \frac{F'}{F} + \frac{n_h V_h^{z_h} + n_r V_r^{z_h}}{n_h V_h} \right)|_{z_h=z_r=z^*} + n_r F V_r \left( \frac{F'}{F} + \frac{n_h V_h^{z_r} + n_r V_r^{z_r}}{n_r V_r} \right)|_{z_h=z_r=z^*} = 0 \qquad (4)$$

where we have suppressed the functional dependencies for ease of presentation. The first term gives the group fitness change due to a marginal increase in 'helper' cooperation, and the second term gives the group fitness change due to a marginal increase in 'reproductive' cooperation. So, at the uniform strategy, $z^*$, any increase in the fitness caused by the increased cooperation of one subgroup of the population is balanced by a commensurate decrease in fitness caused by the same increase in cooperation for the other subgroup $\frac{\partial W(z_h,z_r)}{\partial z_h}|_{z_h=z_r=z^*} = -\frac{\partial W(z_h,z_r)}{\partial z_r}|_{z_h=z_r=z^*}$.

The constrained optimum, $(z_h, z_r) = (z^*, z^*)$, computed using **Equation 4** does not necessarily correspond to a critical point of $W$, that is, the first derivates $\frac{\partial W(z_h,z_r)}{\partial z_h}$ and $\frac{\partial W(z_h,z_r)}{\partial z_r}$ might not vanish at this point. However, if the functions $V_h$ and $V_r$ are symmetrical, in the sense that

$$n_h V_h (z_h, z_r) = n_r V_r (z_r, z_h), \qquad (5)$$

then the fitness function satisfies $W(z_h, z_r) = W(z_r, z_h)$, and it can be seen that this implies that the point $(z_h, z_r) = (z^*, z^*)$ is actually a critical point of $W(z_h, z_r)$.

### Division of labour by between-individual differences

Division of labour by between-individual differences occurs if either of the first two terms of the Taylor expansion (**Equation 2**) is non-zero ($W^{z_h} \neq 0$ or $W^{z_r} \neq 0$). In this case, directional selection will increase or decrease the level of cooperation of one of the individual types. We give here the associated partial differentials of fitness (**Equation 1**):

$$\frac{\partial W(z_h,z_r)}{\partial z_h}|_{z_h=z_r=z^*} = n_h F' V_h + n_h F V_h^{z_h} + n_r F V_r^{z_h}. \qquad (6)$$

$$\frac{\partial W(z_h,z_r)}{\partial z_r}|_{z_h=z_r=z^*} = n_r F' V_r + n_h F V_h^{z_r} + n_r F V_r^{z_r}. \qquad (7)$$

These expressions capture the fitness consequences of a marginal increase in cooperation by helpers and reproductives, respectively. The first term of each captures the fecundity cost to own type of producing more public good, whereas the second term and third term are the viability benefits that accrue to both types from this increased cooperation. If directional selection in both traits is zero ($\frac{\partial W(z_h,z_r)}{\partial z_h}|_{z_h=z_r=z^*} = 0$ and $\frac{\partial W(z_h,z_r)}{\partial z_r}|_{z_h=z_r=z^*} = 0$), then $z_h = z_r = z^*$ is a critical point, and **Equations 6** and **7** imply that

$$\frac{F'}{F}|_{z_h=z_r=z^*} = -\frac{n_h V_h^{z_h}+n_r V_r^{z_h}}{n_h V_h}|_{z_h=z_r=z^*}. \tag{8}$$

$$\frac{F'}{F}|_{z_h=z_r=z^*} = -\frac{n_h V_h^{z_r}+n_r V_r^{z_r}}{n_h V_h}|_{z_h=z_r=z^*}. \tag{9}$$

These equations mean that, if $z_h = z_r = z^*$ is a critical point, then any marginal viability benefit to the group of increased cooperation by one subgroup is cancelled by the fecundity cost to that same subgroup. Moreover, *Equations 8* and *9* together imply that

$$\frac{n_h V_h^{z_h}+n_r V_r^{z_h}}{n_h V_h}|_{z_h=z_r=z^*} = \frac{n_h V_h^{z_r}+n_r V_r^{z_r}}{n_h V_h}|_{z_h=z_r=z^*} \tag{10}$$

If this equation does not hold, then $z_h = z_r = z^*$ is not a critical point, that is, there is a difference in the viability-fecundity tradeoffs between subgroups such that some individuals (without loss of generality, helpers) can secure larger benefits for the group at the same fecundity cost as others (reproductives). This gives our first condition for division of labour being able to evolve:

The between-individual differences condition for division of labour

$$\frac{n_h V_h^{z_h}+n_r V_r^{z_h}}{n_h V_h}|_{z_h=z_r=z^*} > \frac{n_h V_h^{z_r}+n_r V_r^{z_r}}{n_h V_h}|_{z_h=z_r=z^*}. \tag{11}$$

If individuals are indistinguishable when both types invest equally in cooperation ($z_h = z_r = z$), then the viability functions satisfy $V_h(z,z) = V_r(z,z)$. In this case, Condition 11 can be restated as:

$$V_h^{z_h} + \frac{n_r}{n_h} V_r^{z_h} > V_r^{z_r} + \frac{n_h}{n_r} V_h^{z_r}. \tag{12}$$

This says that the contribution to total viability from the increased specialisation of helper individuals is strictly larger than the contribution to total viability from the increased specialisation of reproductives. As a result, helpers are predisposed to become more helper-like as they can gain larger viability gains for the group than the other type of individual.

## Division of labour by an accelerating return from individual specialisation

Division of labour by an accelerating return from individual specialisation can occur if either of the third or fourth terms in the Taylor expansion (*Equation 2*) are positive in value ($W^{z_h z_h} > 0$ or $W^{z_r z_r} > 0$). Taking the second derivative of fitness (*Equation 24*) with respect to each trait and evaluating at the critical point of uniform cooperation $z^*$ gives:

$$\frac{\partial^2 W}{\partial z_h^2}|_{z_h=z_r=z^*} = 2n_h F' V_h^{z_h} + n_h F'' V_h + n_h F V_h^{z_h z_h} + n_r F V_r^{z_h z_h}. \tag{13}$$

$$\frac{\partial^2 W}{\partial z_r^2}|_{z_h=z_r=z^*} = 2n_r F' V_r^{z_r} + n_r F'' V_r + n_h F V_h^{z_r z_r} + n_r F V_r^{z_r z_r}. \tag{14}$$

The terms of *Equations 13 and 14* capture the second-order effects of increased investment in cooperation. The first term of each captures the decline in the fitness benefit of increased cooperation due to the cross-interaction between fecundity and viability. For instance, as a helper invests more in cooperation (higher $x$), it increases its own viability (higher $V_h$), but its fecundity declines as well (lower $F$) and so the relative benefit of this increased viability is lessened (the cross term $F' V_h^{z_h}$ is negative). This represents a kind of decelerating return from cooperation. The second term of each captures the second-order effect of decreased investment in fecundity. If this term is positive, then this means that there is a diminishing fecundity cost to increased investment in cooperation, which can favour division of labour. The third and fourth terms capture the second-order effect of increased investment in viability, that is, does each successive investment in the public good lead to a larger or smaller increase in viability than the previous investment of the same size? The return on investment (ROI) in viability is accelerating if $V_h^{z_h z_h} > 0$, $V_r^{z_h z_h} > 0$, $V_h^{z_r z_r} > 0$, and $V_r^{z_r z_r} > 0$. The ROI is diminishing if these second derivates are negative: $V_h^{z_h z_h} < 0$, $V_r^{z_h z_h} < 0$, $V_h^{z_r z_r} < 0$, and $V_r^{z_r z_r} < 0$.

Thus if either *Equation 13* or *Equation 14* is positive, then division of labour is favoured to evolve. This gives the second condition for division of labour.

The accelerating returns from individual specialisation condition for division of labour

$$\frac{n_h V_h^{z_h z_h} + n_r V_r^{z_h z_h}}{n_h V_h}|_{z_h=z_r=z^*} + \frac{F''}{F}|_{z_h=z_r=z^*} + 2\frac{F'}{F}\frac{V_h^{z_h}}{V_h}|_{z_h=z_r=z^*} > 0, \tag{15}$$

$$\frac{n_h V_h^{z_r z_r} + n_r V_r^{z_r z_r}}{n_r V_r}|_{z_h=z_r=z^*} + \frac{F''}{F}|_{z_h=z_r=z^*} + 2\frac{F'}{F}\frac{V_h^{z_r}}{V_r}|_{z_h=z_r=z^*} > 0. \tag{16}$$

Fixing our attention on just *Equation 15* (or equivalently on 16), this condition states that the sum of the second-order viability effect from increased cooperation (first term on left-hand side) and the second-order fecundity effect of increased cooperation (second term on left-hand side) must be larger than the marginal fecundity cost of increased investment in viability (third term on left-hand side). Note that the third term on left-hand side is always negative because increased investment in viability decreases the value of increased investment in fecundity. Therefore, division of labour by a single-trait mutation can only happen if there is an accelerating ROI in at least one of fecundity, $F(z)$; helper viability, $V_h(z_h, z_r)$; or reproductive viability, $V_r(z_h, z_r)$.

## Division of labour by reciprocal specialisation

The last remaining scenario for division of labour is that the resident strategy of uniform cooperation is unstable to mutations in both traits, which can occur depending on the value of the last term of the Taylor expansion (*Equation 2*; $W^{z_h z_r}\Delta z_h \Delta z_r$). This kind of instability can arise if there is a joint mutation that affects the level of cooperation of both helpers and reproductives at the same time ($\Delta z_h \neq 0$ and $\Delta z_r \neq 0$). However, it could also occur if a slightly deleterious mutation in one trait invades by drift and destabilises the other trait so much that the population evolves away from the critical point. In either case, it will be found that same condition must be satisfied in order for division of labour to evolve. In the rest of this section, we give the general analysis of whether $(z_h, z_r) = (z^*, z^*)$ is unstable to two-trait mutations, and then consider a simplifying special case to clarify the biological interpretation of this analysis.

Suppose that the resident strategy of uniform cooperation ($(z_h, z_r) = (z^*, z^*)$) is stable against single-trait mutations (i.e. Conditions 15 and 16 not satisfied). Then the resident strategy is unstable to two-trait mutations if and only if the determinant of the Hessian is negative:

$$\frac{\partial^2 W}{\partial z_h^2}\frac{\partial^2 W}{\partial z_r^2} < \left(\frac{\partial^2 W}{\partial z_h z_r}\right)^2. \tag{17}$$

This condition is satisfied if the strength of directional selection pushing the population back to the critical point along either of the trait-value directions is less than the strength of directional selection on a trait when moved off of the critical point along the other trait direction. To evaluate the Hessian condition, we first compute the second-order cross derivatives:

$$\frac{\partial^2 W}{\partial z_h z_r}|_{z_h=z_r=z^*} = \frac{\partial^2 W}{\partial z_r z_h}|_{z_h=z_r=z^*} = F'\left(n_h V_h^{z_r} + n_r V_r^{z_r}\right) + F\left(n_h V_h^{z_r z_h} + n_r V_r^{z_r z_h}\right). \tag{18}$$

Here, we have used that $V_i^{z_h z_r} = V_i^{z_r z_h}$. Substituting this, and *Equations 13 and 14*, into the Hessian condition gives:

The reciprocal specialisation condition for division of labour

$$
\begin{aligned}
n_h V_h n_r V_r &\left(\frac{n_h V_h^{z_h z_h} + n_r V_r^{z_h z_h}}{n_h V_h} + \frac{F''}{F} + 2\frac{F'}{F}\frac{V_h^{z_h}}{V_h}\right)\left(\frac{n_h V_h^{z_r z_r} + n_r V_r^{z_r z_r}}{n_r V_r} + \frac{F''}{F} + 2\frac{F'}{F}\frac{V_h^{z_r}}{V_r}\right)|_{z_h = z_r = z^*} \\
&< \left(\left(n_h V_h^{z_r z_h} + n_r V_r^{z_r z_h}\right) + \frac{F'}{F}\left(n_h V_h^{z_r} + n_r V_r^{z_r}\right)\right)^2|_{z_h = z_r = z^*}
\end{aligned} \tag{19}
$$

Assume that neither Condition 15 nor 16 is satisfied, that is, the individual ROI is non-accelerating. Then the left-hand side of the inequality is strictly positive, which means that Condition 19 is nontrivial. We will see in examples that Condition 19 can be satisfied, which means that division of labour can evolve by reciprocal specialisation even when the individual ROI is diminishing.

To clarify further the biological meaning of Condition 19, consider a simple family of models in which viability and fecundity are linear functions. In this case, all second derivates are zero, and so we get the simplified condition:

Simplified reciprocal specialisation condition for division of labour

$$4n_h V_h^{z_h} n_r V_r^{z_r}|_{z_h=z_r=z^*} < \left(n_h V_h^{z_r} + n_r V_r^{z_h}\right)^2|_{z_h=z_r=z^*}. \tag{20}$$

In the case that the viability functions are (*Equation 5*) this condition further simplifies to:

$$V_h^{z_h}|_{z_h=z_r=z*} \; < \; V_h^{z_r}|_{z_h=z_r=z*}. \tag{21}$$

This inequality is satisfied if the viability of reproductives increases faster with increased cooperation from helpers, than it does from increased cooperation from reproductives. This makes clear that reciprocal specialisation can evolve if reproductives stand to gain more from help from helpers than they would gain by helping themselves.

## Acknowledgements

We thank Thomas Scott and Zheren Zhang for their helpful comments and suggestions; St John's College, Oxford (GAC), and the ERC (Horizon 2020 Advanced Grant 834164; SAW) for funding.

## Additional information

### Funding

| Funder | Grant reference number | Author |
|---|---|---|
| H2020 European Research Council | 834164 | Stuart Andrew West |

The funders had no role in study design, data collection and interpretation, or the decision to submit the work for publication.

### Author contributions

Guy Alexander Cooper, Hadleigh Frost, Conceptualization, Formal analysis, Visualization, Writing – original draft, Writing – review and editing; Ming Liu, Stuart Andrew West, Conceptualization, Formal analysis, Writing – original draft, Writing – review and editing

### Author ORCIDs

Guy Alexander Cooper http://orcid.org/0000-0002-1748-8183
Ming Liu http://orcid.org/0000-0002-5170-8688
Stuart Andrew West http://orcid.org/0000-0003-2152-3153

### Decision letter and Author response

Decision letter https://doi.org/10.7554/eLife.71968.sa1
Author response https://doi.org/10.7554/eLife.71968.sa2

## Additional files

### Supplementary files

• Transparent reporting form

### Data availability

The Matlab (R2020b) source code used to generate Figures 4D-F is available at https://osf.io/nw8gz/.

The following dataset was generated:

| Author(s) | Year | Dataset title | Dataset URL | Database and Identifier |
|---|---|---|---|---|
| Cooper GA | 2021 | Does the evolution of division of labour require accelerating returns from individual specialisation? | https://doi.org/10.17605/OSF.IO/NW8GZ | Open Science Framework, 10.17605/OSF.IO/NW8GZ |

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

# Appendix 1

## A Group structure models

We now consider models that have specified spatial (or 'network') structures. We apply the division of labour scenarios outlined in the Materials and methods to study division of labour in these graph models. We find simple, general conditions on the graph for when division of labour can invade.

### A.1 Division of labour by between-individual differences: stars and branching groups

Consider a star of $n$ cells, with 1 focal cell, and $n-1$ peripheral cells (*Appendix 1—figure 1*). Assume that some function, $H(z)$, determines the amount of a shared resource produced by a cell with level of cooperation $z$. And let $1-\lambda$ be the proportion of benefits produced by an individual that it keeps for itself, so that $\lambda$ is the proportion of benefits that is shared equally by a cell's neighbours. If the peripheral cells in the star have phenotype $z_h$ and the focal cell has phenotype $z_r$, then

$$
\begin{aligned}
W(z_h, z_r) &= (n-1)F(z_h)\left((1-\lambda)H(z_h) + \tfrac{\lambda}{n-1}H(z_r)\right)\\
&\quad +F(z_r)\left((1-\lambda)H(z_r) + (n-1)\lambda H(z_h)\right).
\end{aligned}
\tag{22}
$$

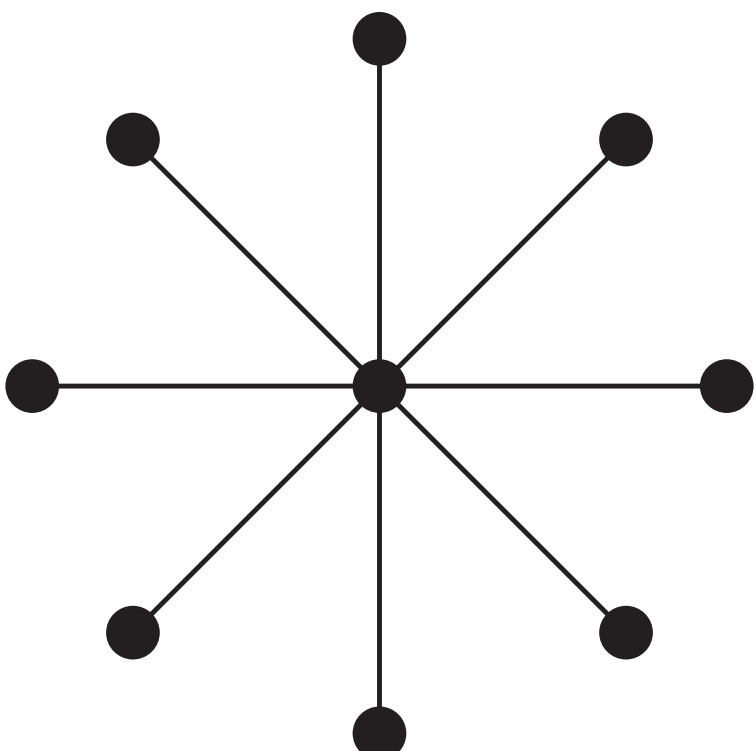

**Appendix 1—figure 1.** A star.

The viability of a peripheral cell is $V_h(z_h, z_r) = (1-\lambda)H(z_h) + \lambda H(z_r)/(n-1)$, and the viability of the focal cell is $V_r(z_h, z_r) = (1-\lambda)H(z_r) + (n-1)\lambda H(z_h)$. *Equation 12* then gives the following condition for division of labour by between-individual differences:

$$
\frac{1}{1 - \frac{n-2}{n-1}\lambda} > \frac{1}{1 + (n-2)\lambda},
\tag{23}
$$

which holds for every positive $\lambda > 0$.

   In a similar vein, consider a group with a branching structure, where alternating cells have either $a$ or $b$ neighbours, with $a < b$. We term the cells with $a$ neighbours as edge cells and cells with $b$ neighbours as node cells. (In order to avoid edge effects, we assume that the group eventually wraps around on itself, creating the shape of a regular polyhedron: in the case of $a = 2$, $b = 3$,

this would be the dodecahedron, with $n_h = 30$ edge cells and $n_r = 20$ node cells.) This gives the following expected fitness:

$$\begin{aligned} W(z_h, z_r) \quad &= n_h F(z_h)\Big((1-\lambda)H(z_h) + (a/b)\lambda H(z_r)\Big) \\ &+ n_r F(z_r)\Big((1-\lambda)H(z_r) + (b/a)\lambda H(z_h)\Big), \end{aligned} \tag{24}$$

where $n_h$ is the number of edge cells and $n_r$ is the number of node cells. Note that for any graph in which $n_h$ cells have degree $a$, and $n_r$ cells have degree $b$, we have that $n_h/n_r = b/a$. The viability of an edge cell is $V_h(z_h, z_r) = (1-\lambda)H(z_h) + (a/b)\lambda H(z_r)$ and the viability of a node cell is $V_r(z_h, z_r) = (1-\lambda)H(z_r) + (b/a)\lambda H(z_h)$. **Equation 12** then gives the following condition for division of labour by between-individual differences:

$$\frac{1}{1 - \frac{b-a}{b}\lambda} > \frac{1}{1 + \frac{b-a}{a}\lambda}. \tag{25}$$

So, since $b > a$, we find that division of labour by between-individual differences occurs for all values of $\lambda > 0$ (any social trait).

In order to calculate the benefit of division of labour in **Figure 4** of the main text, we evaluate $\max(0, W(z^* + \Delta z, z^*) - W(z^*, z^*), W(z^*, z^* - \Delta z) - W(z^*, z^*))$, where we approximate fitness using a first-order Taylor expansion and setting $\Delta z = 0.01$.

## A.2 General graph analysis: between-individual differences

Consider a graph with vertices $i = 1, 2, ..., n$, each vertex with degree $d_i$. Let $A_{ij}$ be the matrix of adjacencies: $A_{ij} = 0$ if there is no edge $i - j$, and for every edge $i - j$ we have $A_{ij} = -1/d_j$. Moreover, $A_{ii} = 1$, for each . Notice that the column sums of the matrix $A$ are zero:

$$\sum_{i=1}^{n} A_{ij} = 0. \tag{26}$$

The fitness function of the group associated with this graph is

$$W = \sum_i F(z_i) \sum_j (\delta_{ij} + A_{ij})H(z_j), \tag{27}$$

where $z_i$ is the level of cooperation of the individual at vertex  , and $\delta_{ij} = 1$ if $i = j$ and 0 otherwise. The uniform cooperation strategy, $z_i = z \,\forall i$, has fitness

$$W = nF(z)H(z), \tag{28}$$

which is maximised for some $z = z^*$ between 0 and 1, by our assumption that $F$ is decreasing in $z$ and $H$ is increasing in $z$.

We now restrict to the 'marginal' case that $F(z)$ and $H(z)$ are both linear functions. In this case we can, without loss of generality, take $F(z) = 1 - z$ and $H(z) = z$. The uniform cooperation strategy is given by $z^* = 1/2$. The first derivatives of $W$ at the uniform strategy are given by:

$$\partial_i W\big|_{z^*} = -\frac{1}{2}\sum_j A_{ij}. \tag{29}$$

Notice that, for fixed $i$,

$$\sum_j A_{ij} = 1 - \sum_{\text{edges } i-j} \frac{1}{d_j}. \tag{30}$$

If all the vertices of the graph have the *same degree*, $d_1 = d_2 = ... = d$, then $\sum_j A_{ij} = 0$, and so the first derivatives all vanish. If one or more of the vertices, $j$, connected to  , does not have the degree $d_i$, then the sum, **Equation 30**, does not vanish. In this case, the uniform strategy is not stable, and division of labour can invade.

For example, consider the branching tree with  seven cells, shown in **Appendix 1—figure 2**. The $A$ matrix of this tree is:

$$A = \begin{bmatrix} 1 & -1/3 & -1/3 & 0 & 0 & 0 & 0 \\ -1/2 & 1 & 0 & -1 & -1 & 0 & 0 \\ -1/2 & 0 & 1 & 0 & 0 & -1 & -1 \\ 0 & -1/3 & 0 & 1 & 0 & 0 & 0 \\ 0 & -1/3 & 0 & 0 & 1 & 0 & 0 \\ 0 & 0 & -1/3 & 0 & 0 & 1 & 0 \\ 0 & 0 & -1/3 & 0 & 0 & 0 & 1 \end{bmatrix} \tag{31}$$

Using *Equation 29*, it follows that

$$\partial_i W \big|_{q^*} = \begin{cases} +1/3 & i = 1 \\ +2/3 & i = 4, 5, 6, 7 \\ -5/2 & i = 2, 3 \end{cases} \tag{32}$$

So vertices 2 and 3 are 'predisposed' towards helping less, whereas the more peripheral vertices ($i = 1, 4, 5, 6, 7$) are predisposed towards helping more. The vertices with the fewest neighbours ($i = 1, 4, 5, 6, 7$) are also the most predisposed towards helping more.

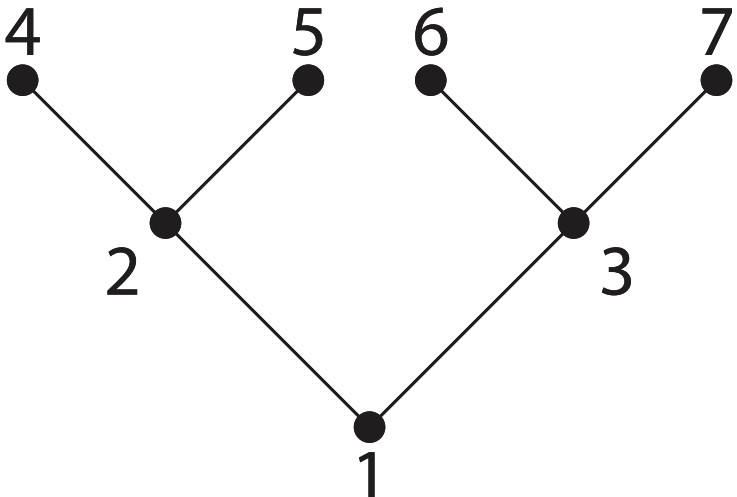

**Appendix 1—figure 2.** A tree.

## A.3 A ring of cells

Let us assume that the group is composed of $n$ cells that form a one-dimensional filament wrapped into a ring, where $n$ is an even number (*Appendix 1—figure 3*). Label the cells 1 through $n$, and assume that 1 is neighbours with $n$. We assume that each cell in the filament shares social benefits with only its direct neighbours. Suppose that 'odd' cells are putative helper cells, and 'even' cells are reproductives. This gives the following fitness for the group:

$$W(z_h, z_r) = \tfrac{n}{2} F(z_h) \left( (1 - \lambda) H(z_h) + \lambda H(z_r) \right) + \tfrac{n}{2} F(z_r) \left( (1 - \lambda) H(z_r) + \lambda H(z_h) \right). \tag{33}$$

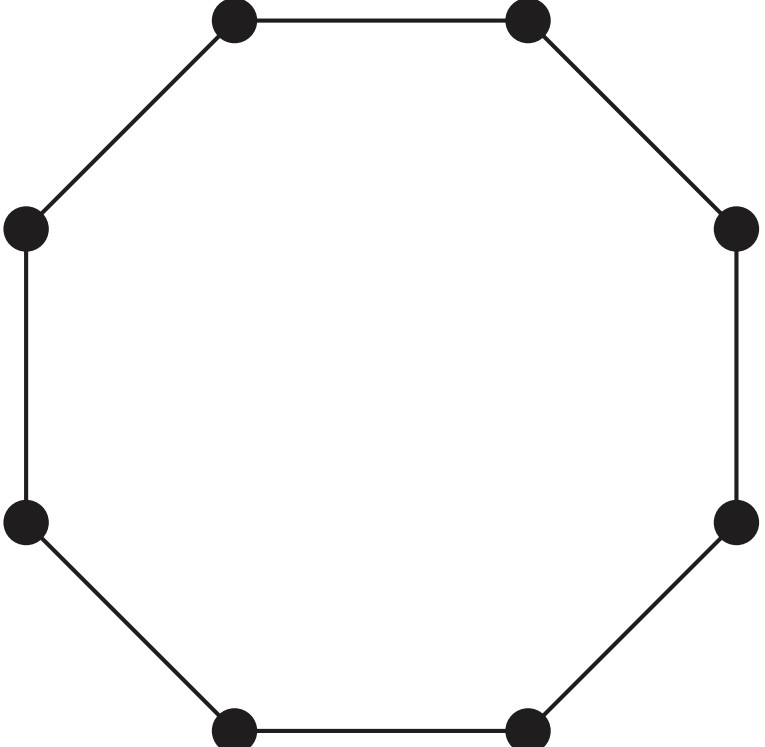

**Appendix 1—figure 3.** A ring of cells.

The viability of helpers is $V_h(z_h, z_r) = (1 - \lambda)H(z_h) + \lambda H(z_r)$ and the viability of reproductives is $V_r(z_h, z_r) = (1 - \lambda)H(z_r) + \lambda H(z_h)$. We now consider each of the possible routes by which division of labour can evolve.

Using *Equations 6* and *7*, we find that $F'/F = -H'/H$ at the resident strategy of uniform cooperation for both helpers and reproductives and therefore that the condition for division of labour by between-individual differences (*Equation 11*) can never hold.

Assume that $F$ is a linear function. If the returns from cooperation are non-increasing ($H'' \leq 0$), then we use Condition 17 (*Figure 5a*), to find that division of labour is favoured if and only if

$$\lambda > \frac{1}{2} - \frac{H''H}{4(H')^2}.$$  (34)

In the case that $H'' = 0$, division of labour is favoured only if $\lambda > 1/2$. If $H''$ is less than zero, then the ROIs are *decelerating*, and $\lambda$ must be larger than 1/2 in order for division of labour to be favoured.

In order to calculate the fitness benefit of division of labour in *Figure 4* of the main text, we evaluate $max_{-\pi \leq \theta \leq \pi} W(z^* + \Delta z \cos(\theta), z^* + \Delta z \sin(\theta)) - W(z^*, z^*)$, where we approximate fitness using a second-order Taylor expansion and setting $\Delta z = 0.01$. In this case, we can show analytically that the direction of mutation giving the maximal increase in fitness satisfies $\theta = \{-\pi/4, 3\pi/4\}$.

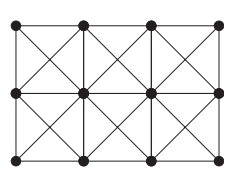

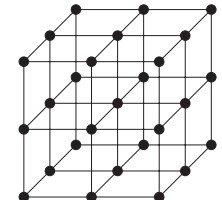

A) $\lambda > \frac{1}{2}$ $(d = 3, \mu = 6)$   B) $\lambda > \frac{1}{2}$ $(d = 4, \mu = 8)$   C) $\lambda > \frac{3}{4}$ $(d = 6, \mu = 8)$

D) Not possible $(d = 8, \mu = 8)$   E) $\lambda > \frac{1}{2}$ $(d = 6, \mu = 12)$

**Appendix 1—figure 4.** Some tessellation patterns, and the associated ranges of $\lambda$ for which reciprocal specialisation is possible in each case. Each figure should be regarded as showing just a few cells of the infinite tesselation.

## A.4 A well-mixed group of cells

We now consider a group that is 'well mixed' such that cells share the benefits of cooperation with all other group members. That is, we let $1 - \lambda$ be the amount of benefits produced by an individual that it keeps for itself and $\lambda$ is the amount of benefits that is shared equally by all other cells in the group. This produces the following expected fitness function:

$$
\begin{aligned}
W(z_h, z_r) &= n_h F(z_h) \left( (1 - \lambda) H(z_h) + \lambda \frac{(n_h - 1)H(z_h) + n_r H(z_r)}{n - 1} \right) \\
&+ n_r F(z_r) \left( (1 - \lambda) H(z_r) + \lambda \frac{(n_r - 1)H(z_r) + n_h H(z_h)}{n - 1} \right),
\end{aligned}
\tag{35}
$$

where $n = n_h + n_r$ and we have that the viability of a helper is $V_h(z_h, z_r) = (1 - \lambda)H(z_h) + \lambda \frac{(n_h - 1)H(z_h) + n_r H(z_r)}{n - 1}$ and the viability of a reproductive is $V_r(z_h, z_r) = (1 - \lambda)H(z_r) + \lambda \frac{(n_r - 1)H(z_r) + n_h H(z_h)}{n - 1}$. Plugging these into **Equation 11**, we find that division of labour cannot evolve by between-individual differences as both sides are equal to $H'/H$, and thus helpers and reproductives face the same viability-fecundity tradeoff.

If the returns from cooperation and fecundity are linear ($F'' = H'' = 0$), then we can evaluate condition 20 directly to find that division of labour by reciprocal specialisation can evolve if (**Figure 5c**)

$$
\lambda > \frac{n - 1}{n}.
\tag{36}
$$

Otherwise, if the returns from cooperation are diminishing (say, $H'' < 0$, with $F$ linear), then we can use the general condition 17 to find,

$$
\lambda > \frac{n - 1}{n} \frac{(1 + n_h x)(1 + n_r x)}{1 + \frac{n_h^2 + n_r^2}{n} x}, \qquad \text{where} \quad x = \frac{H'' F}{2H' F'}.
\tag{37}
$$

Notice that, for diminishing returns ($H'' < 0$), we have that $x > 0$, and in this case it follows that

$$
\frac{(1 + n_h x)(1 + n_r x)}{1 + \frac{n_h^2 + n_r^2}{n} x} > 1.
\tag{38}
$$

In other words, if $H'' < 0$, then $\lambda$ must be *even higher* for division of labour to be favoured, than when $H'' = 0$.

In order to calculate the fitness benefit of division of labour in *Figure 4* of the main text, we evaluate $\max_{-\pi \leq \theta \leq \pi} W(z^* + \Delta z \cos(\theta), z^* + \Delta z \sin(\theta)) - W(z^*, z^*)$, where we approximate fitness using a second-order Taylor expansion and setting $\Delta z = 0.01$.

## A.5 General graph analysis: reciprocal specialisation

We now return to the case of a general graph model, and consider when division of labour is possible via reciprocal specialisation. Once again, fix some graph $G$. Assume that the first derivatives, $\partial_i W$, vanish at the strategy of uniform cooperation ($z_i = z^*$). Recall (*Equation 29*) that this is equivalent to assuming that all the vertices of the graph $G$ have the same degree, $d$. The matrix of second derivatives of $W$ is

$$\text{Hess} = \left. \frac{\partial^2 W}{\partial z_i \partial z_j} \right|_{q^*} = 2F'(z^*)H'(z^*)\left(\delta_{ij} - \lambda M_{ij}\right), \tag{39}$$

where $M$ is a symmetrised version of the $A$ matrix:

$$M_{ij} = \frac{1}{2}(A_{ij} + A_{ji}). \tag{40}$$

All vertices have the same degree, say $d$, so that the matrix $M$ is given by:

$$M_{ij} = \frac{1}{d}L_{ij}, \tag{41}$$

where $L$ is the *Laplacian* matrix:

$$L_{ij} = \begin{cases} d & \text{if } i = j, \\ -1 & \text{if } i - j \text{ is an edge,} \\ 0 & \text{otherwise.} \end{cases} \tag{42}$$

The uniform strategy ($z = z^*$) is *unstable* if and only if the matrix Hess has one or more positive eigenvalues. When $\lambda = 0$, $\text{Hess} = -2I$, where $I$ is the identity. So all eigenvalues of Hess for $\lambda = 0$ are negative, and the uniform strategy is stable. This agrees with the biological intuition: when $\lambda = 0$, the group is not interacting socially, and so division of labour cannot evolve. We would like to find $\lambda^*$ (if it exists), such that division of labour is possible for $\lambda > \lambda^*$. This corresponds to the smallest value of $\lambda$ for which det(Hess) vanishes. We can explicitly evaluate the determinant as:

$$\det(\text{Hess})(\lambda) = \left(\frac{2\lambda}{d}\right)^n \mathcal{P}\left(\frac{d}{\lambda}\right), \tag{43}$$

where the *characteristic polynomial* $\mathcal{P}(x)$ is a degree $n$ polynomial defined by:

$$\mathcal{P}(x) = \det\left(L - xI\right). \tag{44}$$

The roots of $\mathcal{P}(x)$ are all non-negative (this is because $L$ is a symmetric matrix). Moreover, $x = 0$ is a root of $\mathcal{P}(x)$. Let $\mu$ be the largest root of $\mathcal{P}(x)$, that is, $\mu$ is the largest *eigenvalue* of $L$. Then the smallest value of $\lambda$ for which det(Hess)($\lambda$) vanishes is $\lambda^* = d/\mu$. It follows that division of labour can evolve if and only if

$$\lambda\mu > d. \tag{45}$$

Recall that $d$ is the number of neighbours that an individual cell has in the graph. The eigenvalue $\mu$ can be thought of (see below) as a measure of how 'bipartitionable' the graph is. The inequality can then be interpreted loosely as follows: if the graph is made 'more bipartite', $\lambda$ can decrease. If the number of neighbours, $d$, is increased, $\lambda$ must increase.

To further understand $\mu$, a useful property of $L$ is that, for any vector of numbers $x = (x_1, ..., x_N)$:

$$x^T L x = \sum_{i,j=1}^{n} x_i L_{ij} x_j = \sum_{\substack{\text{edges} \\ i-j}} (x_i - x_j)^2. \tag{46}$$

The largest eigenvalue of $L$ is

$$\mu = \max_x \frac{x^T L x}{x^T x}.$$ (47)

On the other hand,

$$\sum_{\substack{\text{edges} \\ i-j}} (x_i - x_j)^2 \leq \sum_{\substack{\text{edges} \\ i-j}} (x_i^2 + x_j^2) = 2d \sum_{i=1}^{n} x_i^2.$$ (48)

It follows that

$$\mu \leq 2d.$$

Moreover, if the graph is bipartite, choose any bipartite colouring of its vertices. Then, assigning $x = +1$ to vertices of one colour, and $x = -1$ to vertices of the other colour, we find that

$$\sum_{\substack{\text{edges} \\ i-j}} (x_i - x_j)^2 = \sum_{\text{edges}} 4 = 2dn.$$ (49)

This means that $\mu$ achieves its maximum value, $\mu = 2d$, for bipartite graphs. It can be shown that $\mu = 2d$ if and only if the graph is bipartite. $\mu$ can be regarded, therefore, as a measure of 'how bipartite' a given graph is. Some representative examples for small ($N \leq 8$) graphs are given in *Appendix 1—figure 4*, *Appendix 1—figure 5*, *Appendix 1—figure 6*, *Appendix 1—figure 7*. The rest of this section applies the above analysis to several basic families of examples.

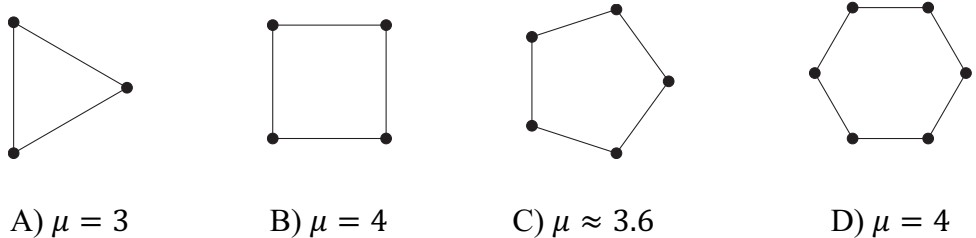

A) $\mu = 3$ B) $\mu = 4$ C) $\mu \approx 3.6$ D) $\mu = 4$

**Appendix 1—figure 5.** Values of $\mu$ for different filament ($d = 2$) graphs.

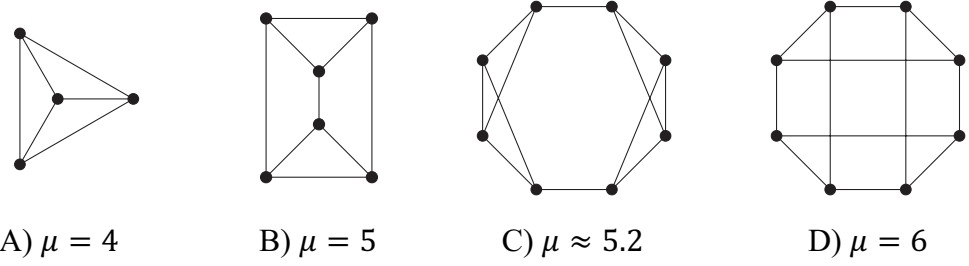

A) $\mu = 4$ B) $\mu = 5$ C) $\mu \approx 5.2$ D) $\mu = 6$

**Appendix 1—figure 6.** Values of $\mu$ for different trivalent ($d = 3$) graphs.

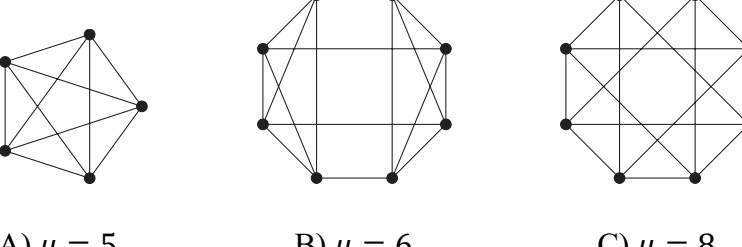

A) $\mu = 5$   B) $\mu = 6$   C) $\mu = 8$

**Appendix 1—figure 7.** Values of $\mu$ for different $d = 4$ graphs.

## Filaments of cells

A ring of $n$ cells has degree $d = 2$, and the largest eigenvalue of the Laplacian matrix depends on whether $N$ is even or odd:

$$\mu = \begin{cases} 4 & n \text{ even} \\ 2 - 2\cos\left(\frac{(n-1)\pi}{n}\right) & n \text{ odd} \end{cases} \tag{50}$$

For $n$ even, division of labour is possible if $\lambda > 1/2$. For large odd $n$, $\mu$ is approximately 4, and so division of labour is again possible for $\lambda > 1/2$. However, for small odd $n$, $\mu$ is less than 4. This means that, for small $n$, division of labour is 'easier' for even numbers of cells than for odd numbers of cells. For some examples, see **Appendix 1—figure 5**.

## Complete graphs

The complete graph with $n$ cells has degree is $d = n - 1$ and the largest eigenvalue is (In terms of **Equation 47**, a possible vector that achieves the maximum is $x_1 = +1$, $x_2 = -1$, and $x_3, x_4, \ldots = 0$. For this vector, $x^T L x = 4 + 2(N - 2) = 2N$ and $x^T x = 2$.)

$$\mu = n. \tag{51}$$

So reciprocal specialisation is possible only if $\lambda > (n - 1)/n$.

## Tessellations

We can carry out similar computations for (infinitely extended) tessellation patterns. There are many bipartite tessellations, in both two and three dimensions. All of these have the maximum possible $\mu$, that is, $\mu = 2d$. Examples of this include the hexagonal lattice, the square lattice, and the 3D cube lattice: **Appendix 1—figure 4(a), (b), and (e)**. Reciprocal specialisation is also possible in many tessellations which are not bipartite. An example with $d = 6$ is shown in **Appendix 1—figure 4(c)**, which has $\mu = 8$. Finally, there are tessellations which are so far from being bipartite that reciprocal specialisation is not possible. An example with $d = 8$ is shown in **Appendix 1—figure 4(d)**, which has $\mu = 8$. The condition $\lambda\mu > d$ cannot be satisfied with $\mu = d$, since $\lambda$ is at most 1.

# B Random role allocation

We repeat the above analyses where we now assume that individuals adopt their phenotypes with a particular probability, independent of other individuals in the group. We find that, with random role allocation, division of labour is not possible (in the absence of accelerating individual ROIs).

## B.1 A ring of cells

Let p be the probability of adopting the role of a helper and investing $x$ in cooperation and $1 - p$ as the probability of adopting the role of a reproductive and investing $y$ in cooperation. In the case of a ring of cells, the number of helpers among a given cell's two neighbours is a binomial random variable, $k \in \{0, 1, 2\}$. This gives the expected fitness:

$$\langle W \rangle = pF(z_h)\Big((1-\lambda)H(z_h) + \lambda \sum_{k=1}^{2} \binom{2}{k}p^k(1-p)^{2-k}(kH(z_h) + (2-k)H(z_r))\Big)$$
$$+(1-p)F(z_r)\Big((1-\lambda)H(z_r) + \lambda \sum_{k=1}^{2} \binom{2}{k}p^k(1-p)^{2-k}(kH(z_h) + (2-k)H(z_r))\Big).$$

(52)

The first term is the expected fitness if the focal cell adopts the role of a helper and the second term is the expected fitness if the focal cell adopts the role of a reproductive. We can simplify the above fitness equation to:

$$\langle W \rangle(z_h, z_r) = pF(z_h)\Big((1-\lambda)H(z_h) + \lambda(pH(z_h) + (1-p)H(z_r))\Big)$$
$$+(1-p)F(z_r)\Big((1-\lambda)H(z_r) + \lambda(pH(z_h) + (1-p)H(z_r))\Big),$$

(53)

If we multiply the fitness equation by $n^2$ and approximate $n_h = pn$ and $n_r = (1-p)n$, then we arrive at the final fitness equation:

$$\langle W \rangle(z_h, z_r) = n_h F(z_h)\Big((1-\lambda)H(z_h)n + \lambda(n_h H(z_h) + n_r H(z_r))\Big)$$
$$+n_r F(z_r)\Big((1-\lambda)H(z_r)n + \lambda(n_h H(z_h) + n_r H(z_r))\Big),$$

(54)

where we now have that the expected viability of a helper is
$V_h(z_h, z_r) = (1-\lambda)H(z_h)n + \lambda(n_h H(z_h) + n_r H(z_r))$ and the expected viability of a reproductive is
$V_r(z_h, z_r) = (1-\lambda)H(z_r)n + \lambda(n_h H(z_h) + n_r H(z_r))$.

The function, *Equation 53*, can be analysed using the results of Section B. We find (as expected) that division of labour by between-individual differences is not possible for this function:

$$V_h^{z_h} + \frac{n_r}{n_h}V_r^{z_h} = nH(z^*) = V_r^{z_r} + \frac{n_h}{n_r}V_h^{z_h}.$$

(55)

Moreover, we can evaluate Condition 20 directly to find that division of labour by reciprocal specialisation can only evolve if (for linear functions $F$ and $H$)

$$(n - \lambda n_r)(n - \lambda n_h) < n_h n_r \lambda^2.$$

(56)

It is impossible to satisfy this inequality for $\lambda \leq 1$. Thus, assuming linear or decelerating ROIs, we find that division of labour cannot evolve by reciprocal specialisation.

In order to calculate the fitness benefit of division of labour in *Figure 4* of the main text, we evaluate $\max_{-\pi \leq \theta \leq \pi, 0 \leq p \leq 1} W(z^* + \Delta z \cos(\theta), z^* + \Delta z \sin(\theta)) - W(z^*, z^*)$, where we approximate fitness using a second-order Taylor expansion and setting $\Delta z = 0.01$. In this case, the above expression must now be maximised over two variables. This is done numerically to produce the main text figure.

## B.2 A branching group of cells

We approximate the fitness of the group as the expected fitness of a randomly chosen cell in the branching group. With probability $n_h/n$ we choose an 'edge cell' with $a$ neighbours and with probability $n_r/n$, we choose a 'node' cell with $b$ neighbours (here $n_h + n_r = n$, $a < b$ and $n_h/n_r = b/a$). If the cell is an edge cell, the number of neighbouring cells that are helpers is a binomial random variable $k \in \{0, \ldots, a\}$. If the cell is a node cell, the number of neighbouring cells that are helpers is a binomial random variable $k \in \{0, \ldots, b\}$. Again, we let p be the probability that the cell adopts the role of a helper. Similar to the process above, we can derive a fitness equation of the form:

$$W(z_h, z_r) = n_h F(z_h)\Big((1-\lambda)H(z_h)n + \Big(\frac{n_h}{n}\frac{a\lambda}{b} + \frac{n_r}{n}\frac{b\lambda}{a}\Big)(n_h H(z_h) + n_r H(z_r))\Big)$$
$$+n_r F(z_r)\Big((1-\lambda)H(z_r)n + \Big(\frac{n_h}{n}\frac{a\lambda}{b} + \frac{n_r}{n}\frac{b\lambda}{a}\Big)(n_h H(z_h) + n_r H(z_r))\Big),$$

(57)

where we have approximated $n_h = pn$ and $n_r = (1-p)n$. We thus have that the viability of a helper is $V_h(z_h, z_r) = (1-\lambda)H(z_h)n + \Big(\frac{n_h}{n}\frac{a\lambda}{b} + \frac{n_r}{n}\frac{b\lambda}{a}\Big)(n_h H(z_h) + n_r H(z_r))$ and the viability of a reproductive is $V_r(z_h, z_r) = (1-\lambda)H(z_r)n + \Big(\frac{n_h}{n}\frac{a\lambda}{b} + \frac{n_r}{n}\frac{b\lambda}{a}\Big)(n_h H(z_h) + n_r H(z_r))$. For the particular case of $a = 2$ and $b = 3$ (a branching filament), we can use *Equation 11* to show that division of labour by between-individual differences can never evolve.

Moreover, if the returns from cooperation and fecundity are linear ($F'' = H'' = 0$), then we can evaluate Condition 20 directly to find that division of labour by reciprocal specialisation can only evolve if $\lambda > 1$, which is outside the physically permissible range of the parameter. Thus, division of labour can only evolve in this scenario if there are individual efficiency benefits from specialisation.

In order to calculate the fitness benefit of division of labour in *Figure 4* of the main text, we evaluate $\max_{-\pi \leq \theta \leq \pi, 0 \leq p \leq 1} W(z^* + \Delta z \cos(\theta), z^* + \Delta z \sin(\theta)) - W(z^*, z^*)$, where we approximate fitness using a second-order Taylor expansion and setting $\Delta z = 0.01$. In this case, the above expression must now be maximised over two variables. This is done numerically to produce the main text figure.

## B.3 General graph analysis: random allocation

In general, suppose that the assignment of phenotype is done randomly, according to a probability density function $\rho(z)$, such that $\int dz \rho(z) = 1$. Write $z_0 = \langle z \rangle$ for the average, and $\sigma^2 = \langle z^2 \rangle - \langle z \rangle^2$ for the variance of the distribution. The expected fitness for an arbitrary graph model (*Equation 27*) is

$$\langle W \rangle = \sum_{i=1}^{n} \int dz_i \rho(z_i) F(z_i) H(z_i) - \lambda \sum_{j=1}^{n} \sum_{i \neq j} \int dz_i dz_j \rho(z_i) \rho(z_j) F(z_i) A_{ij} H(z_j). \tag{58}$$

Putting $F(z) = 1 - z$ and $H(z) = z$, this becomes

$$\langle W \rangle = n \left( z_0 - \sigma^2 - (1 - \lambda) z_0^2 \right), \tag{59}$$

where we have used that

$$\sum_{j=1}^{n} \sum_{i \neq j} A_{ij} = -\sum_{j=1}^{n} A_{jj} = -n. \tag{60}$$

This calculation shows that the expected fitness of the group, $\langle W \rangle$, is independent of the group's graph structure. Moreover, notice that $\langle W \rangle$ is largest (for fixed $z_0$) if the distribution is chosen so that $\sigma^2 = 0$, that is, the state of uniform cooperation. Division of labour would correspond to $\sigma^2 > 0$. So *division of labour cannot be favoured* (absent accelerating returns) if phenotypes are assigned randomly.

To see this in greater detail, consider the distribution

$$\rho(q) = (1 - p)\delta(q - x) + p\delta(z - y). \tag{61}$$

Under this distribution, cells become a helper (phenotype $z_h$) with probability $1 - p$, and a reproductive (phenotype $z_r$), with probability $p$. Then $z_0 = (1 - p)z_h + p z_r$, and

$$\sigma^2 = p(1 - p)(z_h - z_r)^2. \tag{62}$$

The total expected fitness is then

$$\frac{1}{n} < W >= z_0 - p(1 - p)(z - z)^2 - (1 - \lambda)z_0^2. \tag{63}$$

If $\lambda \leq 1/2$, the function has a critical point at $z_h = z_r = z = 1/(2(1 - \lambda))$, where its first derivatives vanish. We compute that the determinant of the Hessian of $\langle W \rangle$ at the critical point is

$$4n^2(1 - \lambda)p(1 - p), \tag{64}$$

which is positive. So

$$z_h = z_r = z = \frac{1}{2(1 - \lambda)} \tag{65}$$

is a maximum of $\langle W \rangle$, for all choices of p, and $\lambda < 1/2$. If $\lambda \geq 1/2$, then $\langle W \rangle$ is maximised for $z_h = z_r = z = 1$. In either case, division of labour is not favoured.

## C Additional questions

### C.1 Does division of labour by reciprocal specialisation produce an accelerating fitness benefit?

We consider a population of uniform cooperators with ESS level of cooperation $z^*$. We consider a mutation where both helpers and reproductives specialise in their respective functions such that $\Delta z_h = -\beta \Delta z_r = \Delta z > 0$, where $\beta > 0$ is the degree to which reproductives specialise more than helpers ($\beta > 1$) or less than helpers ($\beta < 1$). This gives the fitness of a mutant with reciprocal specialisation as:

$$W(z^* + \Delta z, z^* - \beta \Delta z) \approx W(z^*, z^*) + (\frac{1}{2}W^{z_h z_h} + \frac{1}{2}\beta^2 W^{z_r z_r} - \beta W^{z_h z_r})\Delta z^2, \tag{66}$$

where we have assumed that there are no between-individual differences and so $W^{z_h} = W^{z_r} = 0$ at the ESS strategy of uniform cooperation, $z^*$. What is the shape of the return from increased specialisation in this case? We solve for the second derivative of mutant fitness with respect to the degree of specialisation $\Delta z$:

$$\frac{\partial^2 W}{\partial \Delta z^2}(z^* + \Delta z, z^* - \beta \Delta z) \approx (\frac{1}{2}W^{z_h z_h} + \frac{1}{2}\beta^2 W^{z_r z_r} - \beta W^{z_h z_r}), \tag{67}$$

If this expression is positive, then the group fitness returns from reciprocal specialisation are accelerating. In contrast, if this expression is negative, then the group fitness returns from reciprocal specialisation are diminishing.

Let us now assume that the mutant has a higher fitness than the resident strategy of uniform cooperation, such that division of labour by reciprocal specialisation evolves ($W(z^* + \Delta z, z^* - \beta \Delta z) > W(z^*, z^*)$). This means that the second term on the right-hand side of *Equation 66* is strictly positive, which occurs when $\frac{1}{2}W^{z_h z_h} + \frac{1}{2}\beta^2 W^{z_r z_r} - \beta W^{z_h z_r} > 0$. Importantly, this is the exact same condition as for an accelerating fitness return from reciprocal specialisation. Consequently, division of labour by reciprocal specialisation evolves if and only if the group fitness return from reciprocal specialisation is accelerating.

### C.2 Does division of labour without accelerating returns from individual specialisation require that individuals have different viabilities?

In many cases, it makes more sense to consider the viability of the group as a whole rather than the viabilities of the individual cells in the group. For instance, in Volvocine algae, sterile flagella beaters provide a benefit to the group by keeping it afloat at the appropriate height in the water column for light absorption. In this instance, individuals literally sink or swim as a whole group and so all individuals have the same viability. Here, we go through the general framework where we now assume that $V_h = V_r = V$.

Considering *Equation 11*, we find that division of labour by between-individual differences occurs when:

$$\frac{V^{z_h}}{n_h} > \frac{V^{z_r}}{n_r} \tag{68}$$

This specifies that the average contribution by helpers to group viability by a small increase in cooperation is greater than the average contribution by reproductives to group viability by a small increase in cooperation. Thus, putative helpers are predisposed to cooperation.

Let us now assume that there are no between-individual differences ($V^{z_h}/n_h = V^{z_r}/n_r$). Considering *Equations 15 and 16*, we find that division of labour by an accelerating return from individual specialisation can arise if:

$$\left.\frac{nV^{z_h z_h}}{n_h V}\right|_{z_h = z_r = z^*} + \left.\frac{F''}{F}\right|_{z_h = z_r = z^*} + 2\left.\frac{F'}{F}\frac{V^{z_h}}{V}\right|_{z_h = z_r = z^*} > 0 \text{ or} \tag{69}$$

$$\left.\frac{nV^{z_r z_r}}{n_r V}\right|_{z_h = z_r = z^*} + \left.\frac{F''}{F}\right|_{z_h = z_r = z^*} + 2\left.\frac{F'}{F}\frac{V^{z_r}}{V}\right|_{z_h = z_r = z^*} > 0 \tag{70}$$

We thus find that division of labour can evolve if $V^{z_h z_h} > 0$, $V^{z_r z_r} > 0$ or $F'' > 0$. We now assume that there is a non-accelerating return from individual specialisation $V^{z_h z_h} = 0$, $V^{z_r z_r} = 0$ and $F'' = 0$. Considering *Equation 20*, we find that the condition for division of labour by reciprocal

specialisation is: $V^{z_h}/n_h > V^{z_r}/n_r$. However, this contradicts our assumption that there are no between-individual differences ($V^{z_h}/n_h = V^{z_r}/n_r$), and so reciprocal specialisation is not a pathway to division of labour. Consequently, if we assume that all individuals in the group have the same viability ($V_h = V_r = V$), then division of labour evolves by either: (1) between-individual differences or (2) an accelerating return from individual specialisation.

In our main analysis, where different individuals may have different viabilities, the consequence to the fitness of the group when helpers have a lower viability is not modelled. Indeed, a helper produces the same amount of public good for the group regardless of its expected viability. This approximation is also made in the *Yanni et al., 2020*, models and in earlier models by *Michod, 2006*. More realistically, helpers with a lower viability may be expected to produce a smaller total amount of public good over the course of the group life cycle. This in principle could limit the efficiency benefits of between-individual differences and of synergistic reciprocal specialisation. Capturing this feedback between cooperation and viability with a formal model would require additional assumptions, which we leave to future work.

## C.3 What happens if the fitness costs and benefits of cooperation are additive?

In the main analysis, we have assumed that cooperation provides a viability benefit at a personal cost to fecundity, where individual fitness is the product of viability and fecundity. Here, we assume instead that the benefits and costs of cooperation are additive, such that increased cooperation leads to an increase in group fecundity at a cost to personal fecundity. For ease of comparison, we maintain the notation $V_h$ and $V_r$, which now correspond to the fecundity benefit provided to helpers and reproductives by the cooperation of group members.

This gives the following fitness of the group:

$$W(z_h, z_r) = n_h \Big( F(z_h) + V_h(z_h, z_r) \Big) + n_r \Big( F(z_r) + V_r(z_h, z_r) \Big) \tag{71}$$

We can employ the Taylor expansion of fitness (*Equation 2*) to determine the three conditions leading to division of labour. First, we find that division of labour by between-individual differences can arise if:

$$\frac{n_h V_h^{z_h} + n_r V_r^{z_h}}{n_h} > \frac{n_h V_h^{z_r} + n_r V_r^{z_r}}{n_r} \tag{72}$$

This shows that division of labour can evolve if some individuals (helpers) can provide larger fecundity benefits for the group at the same fecundity cost as others (reproductives). Division of labour by an accelerating return from individual specialisation can arise if:

$$\left( n_h F'' + n_h V_h^{z_h z_h} + n_r V_r^{z_h z_h} \right)\Big|_{z_h=z_r=z^*} > 0 \text{ or} \tag{73}$$

$$\left( n_r F'' + n_h V_h^{z_r z_r} + n_r V_r^{z_r z_r} \right)\Big|_{z_h=z_r=z^*} > 0 \tag{74}$$

This shows that division of labour by an accelerating return from individual specialisation can arise if there is an accelerating return from decreased cooperation due to increased personal fecundity $F'' > 0$, or an accelerating return from increased cooperation due to increased viability of the group ($V_h^{z_h z_h}, V_r^{z_h z_h}, V_h^{z_r z_r}, V_r^{z_r z_r} > 0$). If none of the above hold, we find that reciprocal specialisation can evolve if:

$$n_h V_h^{z_h z_r} + n_r V_r^{z_h z_r} < 0, \tag{75}$$

where we have assumed that $\Delta z_h = -\Delta z_r$. This states that there is interference between the cooperative efforts of helpers and reproductives such that increased cooperation by helpers leads to larger relative fecundity benefits for the group when reproductives cooperate less ($V_h^{z_h z_r}, V_r^{z_h z_r} < 0$). There is an analogous effect when fitness is the product of fecundity and viability (first term in small brackets on right-hand side of Condition 17).

