## [Editor Report]

This manuscript presents a theoretical study of the evolution of division of labor, exploring the impact of topology, the convexity and concavity of fitness returns on investment, and different biological modes through which division of labor may arise. This is a difficult topic to study as division of labor evolved long ago, and many theoretical predictions have proven difficult to directly test. The results presented here may provide the next step necessary to produce truly testable hypotheses on how division of labor evolves, and will be of interest to evolutionary biologists, mathematical biologists, and biophysicists.

---

## [Decision Letter]

**Decision letter after peer review:**

Thank you for submitting your article "Does the evolution of division of labour require accelerating returns from individual specialisation?" for consideration by *eLife*. Your article has been reviewed by 2 peer reviewers, and the evaluation has been overseen by a Reviewing Editor and Detlef Weigel as the Senior Editor. The following individual involved in review of your submission has agreed to reveal their identity: Peter J Yunker (Reviewer #2).

Essential revisions:

1) I don't think that the title adequately reflect the novelty of the paper. The work by Rueffler et al. 2012 and Yanni et al. 2020 already provide an answer to this question. We know that accelerating returns are not required. The in-depth analysis on the role of topological constraints and coordination are the key novelties and this should be reflected in the title.

2) Figure 4 and the conclusion on the efficiency benefits at the group level (lines 355-357) are central to the argumentation of the paper. However, the actual group benefits as a function of λ and individual return functions are nowhere shown in the paper. I think it would be very beneficial if the authors add additional panels to Figure 4, perhaps in the form of heat maps, showing how the group benefit is affected by the two main parameters altered.

3) The contradiction with Yanni et al. 2020 boils down to whether λ > (n-1)/n is a realistic scenario or not. Can the authors provide biological examples where this condition could be fulfilled?

4) The findings on lines 431-434 are very important, but the data is not presented in a visual form. I recommend to add an additional figure displaying the actual results, again together with group productivity effects as mentioned in my comment 3. above.

5) It is shown that division of labor can evolve with concave returns on investment if an optimal pattern is produced. Further, it is shown that if specialists are randomly distributed on a graph, specialization can only evolve if returns on investment are convex. This is a very nice result. However, while I really like this result, the discussion of it overstates what has been learned. In particular, perfect realizations of patterning and completely random distributions of specialists represent the two extreme cases. While the authors are likely correct that mechanisms that perfectly reproduce patterns are unlikely to evolve prior to division of labor, mechanisms that produce patterns with random errors very well are different. Single celled organisms have many mechanisms through which they modify behavior based on interactions with their environment and each other. Further, alternating patterns are common in biology and physics, and can emerge from simple low-level rules. So, the possibility of producing a pattern of alternating specialists – with errors – is plausible. The question then becomes, for a given topological structure, how precise must the pattern be reproduced? If perfect (or nearly perfect) patterning is required, then concave specialization via topology seems unlikely. However, if *nearly* random patterns favor concave specialization, this barrier would be significantly reduced. The 'error tolerance' of a given topological structure seems likely to depend on its fitness advantage over generalists. While a thorough analysis of this question would be fantastic to see, I do not think it is necessary for publication. Instead, I think it would be sufficient to expand the discussion of this topic to reflect these uncertainties.

In a related vein, "between individual differences" share some similarities with these imperfect patterns. It is unlikely that between individual differences would arise via sophisticated intercellular communication/interactions. But simple mechanisms, such as stochastic switches or persistent environmental interactions can lead to phenotypic differences. If these phenotypic differences arise via spatial interactions, then a pattern emerges as well.

6) I wonder if the way sharing is modeled here is too generous. If a cell shares all of its viability, shouldn't it be dead? Either (1) the viability that is being shared is not all of that individual's viability, in which case the fitness written is not the actual fitness of the organism, (2) this sharing is a one-time act after which the altruistic cell is dead, (3) or there is some other reason why this scenario is reasonable which needs to be further explained.

7) Line 408 These results do not seem contradictory to me. Different assumptions produce different models, which, in turn, produce different results. For example, if I construct a model with no sharing, division of labor will not evolve, even for accelerating returns. This does not contradict models that include sharing, but is the result of a different assumption / scenario. It can, of course, be discussed which approach is applicable in different scenarios, and which may be more broadly applicable.

8) Line 453 "However, the biological plausibility of any mechanism based on pre-existing cues would need to be explicitly justified and modelled…"

Line 458 "further studies-such as ancestral-state reconstructions-are needed to show whether coordination preceded division of labour in individual species"

These are true statements. However, similarly true statements can be made about nearly any mechanism that facilitates the evolution of division of labor. For example, accelerating returns. Placing all of these mechanisms within this context, or presenting the evidence that exists for some but not others, would be helpful.

9) Equation 3 was another very nice result. However, I think the discussion here could also be clarified. This model appears likely to hold as long as the standard deviation of d is small compared to the mean d. Consider an organism that has 1000 somatic cells that each only connect to one central germ cell; consider also a linear return on investment. Equation 3 suggests that λ must be greater than 0.5. In this absurd case, specialists should produce a higher fitness than generalists even if λ is less than 0.5. Again, while additional work determining the exact limits in which Equation 3 applies would be nice, I do not think they are necessary. Instead, appropriate caveats could be added to this effect.

*Reviewer #1:*

The paper consists of two parts.

The first part deals with "the three pathways to division of labour". It builds on a mathematical model showing that division of labour can evolve when (1) there are accelerating returns for individuals from specialisation, (2) there are pre-existing differences between individuals such that some individuals are predisposed for one task or the other, and (3) there is reciprocal specialisation leading to synergistic efficiency benefits at the group level. This model recovers the findings by Rueffler et al. 2012. This part is very well written and reads more like a review, bringing specialised and non-specialised readers onto the same page.

The second part focusses on pathways (2) and (3), where individual returns can be diminishing, and thus the evolution of division of labour is more difficult to explain. This part is motivated by a recent paper by Yanni et al. 2020 in *eLife* showing that topological constraints (e.g. small network of individuals, limited number of neighbours) are essential to favour division of labour with diminishing returns. The authors challenge this view and provide an in-depth analysis on topological constraints. They show that such constraints in combination with pre-existing differences between individuals and reciprocal specialisation are indeed conducive for the evolution of division of labour, but not essential. They show that when the benefit of cooperation is larger for neighbours than for the co-operator than division of labour can evolve with diminishing returns even in the absence of topological constraints. This is a key new insight. But perhaps even more important, the authors highlight that pathways (2) and (3) rely on the assumption that individuals have access to information from neighbours to coordinate their actions at the group level. The authors show that division of labour cannot evolve with diminishing returns when such information is absent. And because mechanisms of information collection and coordination are likely to only evolve once division of labour is already in place, the authors argue that topological constraints might play a minor role in driving the initial evolutionary steps towards division of labour.

In brief, this is a very insightful paper and significantly advances our theoretical and conceptual understanding of division of labour. It will spur future theoretical and empirical work in the field, and for the latter, the authors present guidelines of how to test the theory.

*Reviewer #2:*

This is a very interesting paper on the evolution of division of labor. In particular, the authors explore the impact of topology, the convexity and concavity of fitness returns on investment, and different biological 'modalities' through which division of labor may arise. This is a difficult topic to study as, in most lineages, division of labor evolved long ago, and thus cannot be directly studied in the lab. Further, many theoretical predictions have proven difficult to directly test. This manuscript furthers our understanding of the underlying theory of the evolution of division of labor, and presents a means to test which modality is responsible for the emergence of division of labor in different cases.

However, there are a few caveats worth mentioning. Comparisons to previous works are not always clear. Different models built with different assumptions can produce different predictions; however, that does not mean they disagree, only that they describe different scenarios. Further, the model used here allows an entity to give away all of its 'viability,' making it unclear how it continues to live and function. Finally, the order in which division of labor and 'patterning' evolve is presented as definitive, when it is ultimately a postulate.

---

## [Author Response]

Essential revisions:1) I don't think that the title adequately reflect the novelty of the paper. The work by Rueffler et al. 2012 and Yanni et al. 2020 already provide an answer to this question. We know that accelerating returns are not required. The in-depth analysis on the role of topological constraints and coordination are the key novelties and this should be reflected in the title.

We have changed the title. There are several novel results in the analysis and so we have chosen a title that we think captures its scope, without focus on any one particular result or use of any jargon.

2) Figure 4 and the conclusion on the efficiency benefits at the group level (lines 355-357) are central to the argumentation of the paper. However, the actual group benefits as a function of λ and individual return functions are nowhere shown in the paper. I think it would be very beneficial if the authors add additional panels to Figure 4, perhaps in the form of heat maps, showing how the group benefit is affected by the two main parameters altered.

This is a really nice idea. We have replaced the relevant panels in Figure 4 with heat maps as suggested. This now shows both when division of labour is favoured to invade (non-white shading) and the relative fitness benefit of division of labour when it does (relative darkness of shading).

3) The contradiction with Yanni et al. 2020 boils down to whether λ > (n-1)/n is a realistic scenario or not. Can the authors provide biological examples where this condition could be fulfilled?

We have expanded discussion of this point, highlighting why there are differences in the predictions of each model, and giving biological examples of cooperation that benefits neighbours more than producers. Lines 420-438.

4) The findings on lines 431-434 are very important, but the data is not presented in a visual form. I recommend to add an additional figure displaying the actual results, again together with group productivity effects as mentioned in my comment 3. above.

Thank you for the suggestion, we have expanded Figure 4 to include new panels that show when division of labour is favoured to evolve under fully random specialisation. These illustrate that an accelerating return form specialisation is required in all three models.

5) It is shown that division of labor can evolve with concave returns on investment if an optimal pattern is produced. Further, it is shown that if specialists are randomly distributed on a graph, specialization can only evolve if returns on investment are convex. This is a very nice result. However, while I really like this result, the discussion of it overstates what has been learned. In particular, perfect realizations of patterning and completely random distributions of specialists represent the two extreme cases. While the authors are likely correct that mechanisms that perfectly reproduce patterns are unlikely to evolve prior to division of labor, mechanisms that produce patterns with random errors very well are different. Single celled organisms have many mechanisms through which they modify behavior based on interactions with their environment and each other. Further, alternating patterns are common in biology and physics, and can emerge from simple low-level rules. So, the possibility of producing a pattern of alternating specialists – with errors – is plausible. The question then becomes, for a given topological structure, how precise must the pattern be reproduced? If perfect (or nearly perfect) patterning is required, then concave specialization via topology seems unlikely. However, if nearly random patterns favor concave specialization, this barrier would be significantly reduced. The 'error tolerance' of a given topological structure seems likely to depend on its fitness advantage over generalists. While a thorough analysis of this question would be fantastic to see, I do not think it is necessary for publication. Instead, I think it would be sufficient to expand the discussion of this topic to reflect these uncertainties.

We have expanded the discussion in the section to highlight that a perfect allocation of labour may not in principle be required, but simply that cell specialisation cannot be fully random (Lines 461-466).

In a related vein, "between individual differences" share some similarities with these imperfect patterns. It is unlikely that between individual differences would arise via sophisticated intercellular communication/interactions. But simple mechanisms, such as stochastic switches or persistent environmental interactions can lead to phenotypic differences. If these phenotypic differences arise via spatial interactions, then a pattern emerges as well.

We have expanded the discussion of this point to include the possibility that the mechanism that produces between individual differences may be co-opted to coordinate division of labour (Lines 484486).

6) I wonder if the way sharing is modeled here is too generous. If a cell shares all of its viability, shouldn't it be dead? Either (1) the viability that is being shared is not all of that individual's viability, in which case the fitness written is not the actual fitness of the organism, (2) this sharing is a one-time act after which the altruistic cell is dead, (3) or there is some other reason why this scenario is reasonable which needs to be further explained.

This is a good point, which we had discussed in supplementary section C.2. Our assumptions align with scenario (2) above and are consistent with the modelling assumptions of the Yanni et al. (2020) paper and Michod (2006*)* paper. We have added a description of this assumption in the main text (Lines 135138) and expanded the discussion in the supplementary information (section C.2).

7) Line 408 These results do not seem contradictory to me. Different assumptions produce different models, which, in turn, produce different results. For example, if I construct a model with no sharing, division of labor will not evolve, even for accelerating returns. This does not contradict models that include sharing, but is the result of a different assumption / scenario. It can, of course, be discussed which approach is applicable in different scenarios, and which may be more broadly applicable.

We have expanded discussion of the assumptions made in the different models. Please see our response to revision point (3) above.

8) Line 453 "However, the biological plausibility of any mechanism based on pre-existing cues would need to be explicitly justified and modelled…"Line 458 "further studies-such as ancestral-state reconstructions-are needed to show whether coordination preceded division of labour in individual species"These are true statements. However, similarly true statements can be made about nearly any mechanism that facilitates the evolution of division of labor. For example, accelerating returns. Placing all of these mechanisms within this context, or presenting the evidence that exists for some but not others, would be helpful.

We have added a paragraph to highlight why we think that a larger burden of evidence is needed for between-cell coordination then for accelerating returns from individual specialisation (Lines 495-500). In short, coordination requires an adaptive argument for why such between-cell behaviour will have evolved prior to or concurrently with the emergence of division of labour. In contrast, the shape of the return from more investment is simply an inherent aspect of the biotic or abiotic environment and so does not require an adaptive justification.

9) Equation 3 was another very nice result. However, I think the discussion here could also be clarified. This model appears likely to hold as long as the standard deviation of d is small compared to the mean d. Consider an organism that has 1000 somatic cells that each only connect to one central germ cell; consider also a linear return on investment. Equation 3 suggests that λ must be greater than 0.5. In this absurd case, specialists should produce a higher fitness than generalists even if λ is less than 0.5. Again, while additional work determining the exact limits in which Equation 3 applies would be nice, I do not think they are necessary. Instead, appropriate caveats could be added to this effect.

We have added text to clarify this point (Lines 332-336). Equation 3 is arrived at by assuming that all cells have the same number of neighbours (standard deviation of d is zero). If this is not the case, then different cells have different numbers of neighbours and therefore division of labour by between individual differences will evolve (as established in the section just prior to this one). If that is the case, the second order analysis that goes into deriving Equation 3 is inconsequential to whether division of labour can evolve.